# PET-CT in Clinical Adult Oncology—VI. Primary Cutaneous Cancer, Sarcomas and Neuroendocrine Tumors

**DOI:** 10.3390/cancers14122835

**Published:** 2022-06-08

**Authors:** Gabriel C. Fine, Matthew F. Covington, Bhasker R. Koppula, Ahmed Ebada Salem, Richard H. Wiggins, John M. Hoffman, Kathryn A. Morton

**Affiliations:** 1Department of Radiology and Imaging Sciences, University of Utah, Salt Lake City, UT 84132, USA; gabriel.fine@hsc.utah.edu (G.C.F.); matthew.covington@hsc.utah.edu (M.F.C.); bhasker.koppula@hsc.utah.edu (B.R.K.); ahmed.salem@utah.edu (A.E.S.); Richard.Wiggins@hsc.utah.edu (R.H.W.); john.hoffman@hci.utah.edu (J.M.H.); 2Faculty of Medicine, Department of Radiodiagnosis and Intervention, Alexandria University, Alexandria 21526, Egypt; 3Intermountain Healthcare Hospitals, Summit Physician Specialists, Murray, UT 84123, USA

**Keywords:** PET, FDG, DOTA-TATE, melanoma, Merkel cell carcinoma, cutaneous squamous cell carcinoma, immune-related adverse events, carcinoid, pancreatic neuroendocrine tumor, paraganglioma

## Abstract

**Simple Summary:**

Positron emission tomography (PET), typically combined with computed tomography (CT), has become a critical advanced imaging technique in oncology. With PET-CT, a radioactive molecule (radiotracer) is injected into the bloodstream and localizes to tumor sites because of specific cellular features of the tumor that accumulate the targeting radiotracer. The CT scan is performed at the same time, facilitating better visualization of radioactivity from deep or dense structures and providing detailed anatomic information. PET-CT has a variety of applications in oncology, including staging, therapeutic response assessment, restaging and surveillance. This series of six articles provides an overview of the value, applications, imaging and interpretive strategies of PET-CT in the more common adult malignancies. In the current and sixth report in this series, the application of PET-CT is reviewed for more aggressive skin cancers, sarcomas and neuroendocrine tumors.

**Abstract:**

PET-CT is an advanced imaging modality with many oncologic applications, including staging, therapeutic assessment, restaging and surveillance for recurrence. The goal of this series of six review articles is to provide practical information to providers and imaging professionals regarding the best use of PET-CT for specific oncologic indications, the potential pitfalls and nuances that characterize these applications, and guidelines for image interpretation. Tumor-specific clinical information and representative PET-CT images are provided. The current, sixth article in this series addresses PET-CT in an evaluation of aggressive cutaneous malignancies, sarcomas and neuroendocrine tumors. A discussion of the role of FDG PET for all types of tumors in these categories is beyond the scope of this review. Rather, this article focuses on the most common malignancies in adult patients encountered in clinical practice. It also focuses on Food and Drug Agency (FDA)-approved and clinically available radiopharmaceuticals rather than research tracers or those requiring a local cyclotron. This information will serve as a guide to primary providers for the appropriate role of PET-CT in managing patients with cutaneous malignancies, sarcomas and neuroendocrine tumors. The nuances of PET-CT interpretation as a practical guide for imaging providers, including radiologists, nuclear medicine physicians and their trainees, are also addressed.

## 1. Introduction

This series of six review articles addresses the role of combined positron emission tomography–computed tomography (PET-CT) in adult clinical oncology. The sixth article in this series focuses on [^18^F]Fluoro-2-deoxy-2-D-glucose (FDG) PET-CT imaging in the management of the more aggressive non-lymphomatous dermal malignancies, including melanoma, Merkel cell carcinoma and cutaneous squamous cell carcinoma. The report also discusses the use of FDG PET-CT in some of the most common sarcomas encountered in clinical practice in adult patients. Finally, the article addresses PET-CT imaging in neuroendocrine tumors and includes both FDG PET-CT and somatostatin receptor (SSTR)-binding octreotide analogs, including NETSPOT^®^ ([68]Ga-DOTA-TATE, AAA Novartis, NY, USA) and Detectnet^®^ ([^64^Cu]Cu-DOTA-TATE, Curium US, LLC, Maryland Heights, MO, USA). These radiopharmaceuticals, combined with widespread availability, technological improvements in instrumentation, such as digital PET-CT, and acquired global experience, have placed PET-CT at the forefront of oncologic imaging for these tumors.

The goal of this review is to provide practical information for referring providers, radiologists, nuclear medicine practitioners and their trainees regarding the best use of PET-CT for primary cutaneous malignancies, sarcomas and neuroendocrine tumors, and the potential pitfalls and nuances that characterize PET-CT imaging in these applications. Hundreds of different types of tumors exist, both pediatric and adult. A discussion of the role of FDG PET-CT for all of these is beyond the scope of a single review article. Rather, this series of reviews focuses on the most common adult malignancies of the aforementioned types that may be encountered in clinical practice. The articles also focus on FDA-approved and clinically accessible radiopharmaceuticals rather than research tracers or those that require a local cyclotron. It is also acknowledged that a large body of literature relates PET characteristics to prognostic factors for specific cancers. This is not directly addressed, as it arguably rarely impacts treatment guidelines and patient care algorithms. Since the use of PET–magnetic resonance imaging (MRI) scanners is limited in the US, the focus of this article is on PET-CT because of its widespread availability. However, basic principles described are also applicable to PET-MRI. The targeted readers for this review are imaging providers, including radiologists, nuclear medicine physicians and their trainees. For these providers, practical guidance is provided, with multiple examples, to optimize image interpretation. The information is also important for medical and surgical care professionals caring for or treating adult cancer patients in order for them to have realistic expectations and a clear understanding of the indications and advantages of PET-CT for their patients.

## 2. Primary Cutaneous Malignancies

### 2.1. Background and General Considerations

The three most common non-lymphomatous aggressive cutaneous malignancies in which FDG PET-CT plays a role are malignant melanoma, cutaneous squamous cell carcinoma and Merkel cell carcinoma [1,2,3]. Invasive basal cell carcinoma is an additional type of skin cancer that can be serious if advanced. However, its morbidity is in local invasion and FDG PET-CT is not typically employed in its management. In evaluating aggressive cutaneous malignancies, FDG PET-CT is not employed in the identification or characterization of the primary lesion, but rather in assessing the presence of distant metastatic disease or its response to treatment. Since metastases can be widespread, whole-body (head-to-toe) FDG PET-CT is typically recommended, although the added value of extended imaging, as opposed to the standard eyes-to-thighs approach, is somewhat controversial [4].

There are a number of considerations that are relevant to all three types of aggressive cutaneous malignancies. Hypermetabolic skin lesions may be difficult to appreciate on FDG PET-CT images if thin. Review of non-attenuation-corrected (NAC) PET images can increase sensitivity for small neoplastic cutaneous lesions. Focal metabolic activity on FDG PET in the skin can result not only from skin cancer, but also from any infectious or inflammatory lesion of the skin, as well as skin trauma and active scar formation. Evaluation of a surgical bed for residual tumor is often difficult with FDG PET-CT. After surgery, scar tissue may remain metabolically active for extended intervals of time, even years. The depth of known skin cancers cannot be well-assessed by FDG PET-CT and requires determination by surgical pathology. 

### 2.2. Challenges Related to Immunotherapy

Many advanced cutaneous malignancies now employ immunotherapy, typically with checkpoint inhibitors, for metastatic disease or for adjuvant control in higher risk patients [5,6,7,8]. These are often used in combination or in conjunction with chemotherapy or other types of targeted therapies, such as anti-programmed cell death protein (PD)-1 antibodies or mitogen-activated protein kinase enzymes (MEKs) and BRAF inhibitors. The development of systemic inflammatory immune-related adverse events (irAEs) related to these immunotherapies, especially when used in combination, results in a wide spectrum of conditions that are FDG PET-CT-positive and can be mistaken for tumor progression. These include a sarcoid-like lymphadenitis, hypophysitis, cerebritis, focal or diffuse lung opacities due to pneumonitis, hepatitis, pancreatitis, colitis and the involvement of many other organ systems (Figure 1) [9,10,11,12,13]. Typically, the known sites of tumor involvement may improve on FDG PET with immunotherapy. However, the development of systemic sites of new FDG activity may signify possible irAEs. It is critical that possible irAEs be recognized not only to avoid the misdiagnosis of tumor progression, but also to avoid serious drug-related toxicities due to irAEs. This principal is important not only in evaluated cutaneous malignancies, but also in a myriad of other tumors for which immunotherapy is now utilized.

An additional consideration related to treatment with immune-modulating drugs is the concept of pseudo-progression. This occurs most commonly with interim PET-CT scans obtained during therapy [14]. The finding of pseudo-progression is the development of a relative increase in metabolic activity in known sites of tumor involvement that is likely due to acute hypoxia of tumors resulting from the effects of immunotherapy on the tumor microvasculature. This is most common in melanoma but can occur with other tumors treated with immunotherapy as well. This should be considered in patients in whom there is a clinical improvement in the face of worsening findings with FDG PET-CT. In pseudo-progression, a repeat delayed FDG PET-CT after, for example, an additional 2-month delay will typically demonstrate a clear improvement. The phenomenon of pseudo-progression is also a significant problem in predicting response to checkpoint inhibitors by an early “interim” scan during treatment [15]. Guidelines have been published regarding the assessment and reporting of FDG PET-CT in cancer patients who have received treatment with immune checkpoint inhibitors [16]. Another variation on pseudo-progression is shown by tumors that are responding to treatment but which increase in size or even number due to edema or hemorrhage in the tumors. This process typically occurs 4–6 weeks after initiation of treatment but can extend for several months. Often, despite an increase in size, FDG activity will decrease in a tumor that is responding to treatment. In some cases, a dissociated response can occur, with some tumors appearing to progress and others responding [16]. Clinical trials utilize a variety of tumor response criteria for assessing treatment effect. Of note, withdrawal of immunotherapy can result in a marked progression of the tumor (“hyper-progression”) [17]. As recommended by the consensus guidelines [16], it is critical that a baseline FDG PET-CT be performed prior to initiation of treatment. An interim scan at 8–12 weeks of therapy may be helpful to resolve discordant findings with CT. If there is clinical deterioration, an earlier scan can be performed. Finally, a scan should be performed prior to discontinuation of immune checkpoint inhibitors so that accurate assessment of subsequent progression can be made.

Despite some diagnostic uncertainty due to systemic inflammatory response to immunotherapy, FDG PET-CT has been shown to be of significant value in determining response to treatment for melanoma [18]. Increasingly, therapeutic response criteria are increasingly incorporating changes in metabolic activity to judge therapeutic response and clinical benefit. These include the PERCIST criteria as well as formal incorporation of FDG PET parameters into traditional anatomically based response criteria, such as RECIST, PERCIST, EORTC, PETCRIT and PERCIMT [19]. 

### 2.3. Melanoma

Melanoma is an aggressive melanocytic neoplasm that is increasing in frequency and is currently the fifth most common cancer in both men and women. Melanoma is broadly classified as epithelial (cutaneous) or non-epithelial (non-cutaneous). Non-epithelial melanomas include mucosal melanoma (vulvar, vaginal, oral cavity, anorectal, esophageal, etc.) and ocular melanoma (uveal, conjunctival). There are multiple categories and subtypes based on a revised World Health Organization (WHO) classification that depend primarily upon epidemiological, clinical, morphologic and genomic features [20]. Melanoma can also be classified as being due to cumulative sun exposure (CSE) or not. Most cutaneous melanomas are due to CSE, whereas mucosal and ocular melanomas are not typically related to CSE. 

This section includes discussions of the role of FDG PET-CT in the management of melanomas both of cutaneous and non-cutaneous origin. Non-cutaneous melanomas comprise only 7% of melanomas and 75% of non-cutaneous melanomas are ocular, with 25% being mucosal. Non-cutaneous melanomas have a worse prognosis than cutaneous melanomas, with a 5-year survival of approximately 28% compared to approximately 76% for cutaneous melanomas [21]. Prognosis depends upon the stage and the primary site of disease, but not, as for cutaneous lymphoma, on the depth of the primary tumor. Melanomas involving unusual sites tend to have worse prognoses.

#### 2.3.1. Cutaneous Melanoma

Cutaneous melanoma accounts for the minority of skin cancers but the majority of skin cancer-related deaths [22]. Cutaneous melanoma has markedly increased in frequency, with survival in most cases dependent upon the depth of the lesion (Breslow depth) [23]. Treatment of cutaneous melanoma is typically by Mohs surgery and wide excision, the latter often combined with sentinel lymph node biopsy for lesions greater than 1 mm depth or those with high-risk histologic features, regardless of depth. Pembrolizumab is now FDA-approved for adjuvant treatment of stage IIB or C tumors that have been completely resected. For first-line treatment of widely metastatic disease, combination targeted and immunotherapeutic approaches are typically utilized. A number of active clinical therapeutic trials exist.

Staging of cutaneous melanomas is performed with a sentinel lymph node biopsy (SLNB) for primary lesions with a Breslow thickness of >0.8–1.0 mm or <0.8 mm with ulceration. However, false-negative sentinel lymph node identification by scintigraphy can occur in older patients, who have poor lymphatic clearance of injected tracers, and those with ulcerated or thick primary lesions [24]. FDG PET-CT as well as high-resolution ultrasound have not been shown to be effective in preoperative distinction between sentinel lymph nodes that are positive for tumors and those that are negative [25]. The yield of PET-CT in sentinel lymph node-negative cutaneous melanoma is low, but the value of FDG PET-CT in staging and defining the extent of advanced or metastatic disease is well-established (Figure 2) [26]. Metastatic cutaneous melanoma is typically intensely hypermetabolic (Figure 3). The National Comprehensive Cancer Network (NCCN) endorses the option of FDG PET-CT at the outset for stage III or IV disease, acknowledging that it may be more sensitive than other modalities, particularly for disease in the extremities (Figure 4) [27]. Typically, patients are scanned from the top of the head through the feet because metastatic disease can follow unpredictable pathways and occurs widely. For stage I or II disease, FDG PET-CT has a sensitivity of 0–67% and a specificity of 77–100% and it is not typically recommended except in problem-solving situations [28]. This is compared to a sensitivity of 68–87% and a specificity of 92–98% for stage III and IV disease [28]. 

Despite the limitations of FDG PET-CT in assessing response to immunotherapy, FDG PET-CT at more delayed time points (e.g., 10 months post-treatment) outperforms CT and MRI in assessing complete response [18,29]. The use of FDG PET-CT in assessing treatment response should become more uniformly applied with the formal incorporation of changes in metabolic activity into structured response criteria, as discussed above. 

#### 2.3.2. Ocular Melanoma

Ocular melanoma involves the eye and is a distinct category of melanoma. Ocular melanoma can involve the conjunctiva or uvea. The uvea, the site of most ocular melanomas, consists of the iris, ciliary body and choroid. The most common uveal site is the choroid (posterior uvea). The most frequent site of metastatic disease spread in choroidal melanomas, which unfortunately occurs early, is the liver. The primary site of uveal melanoma may be difficult to appreciate with FDG PET-CT images, which is not simply a function of tumor size [30,31]. Distant metastases with ocular melanoma can differ in magnitude of uptake, some being characterized by little or no metabolic activity (Figure 5 and Figure 6) [32]. MRI shows better performance than PET-CT for detecting small liver metastases due to uveal melanoma [33]. Treatment of posterior uveal melanoma is typically by visual-sparing mechanisms, including watchful waiting for older patients, plaque brachytherapy for smaller lesions and enucleation for larger lesions, followed by cryotherapy of residual soft tissues [34,35]. Systemic metastases to the liver are typically managed by liver-directed therapy. Systemic cytotoxic chemotherapy for metastatic disease has been largely unsuccessful. Targeted therapy and immunotherapy approaches are not as well-developed for ocular melanoma as they are for cutaneous melanoma but are under development—T-cell-directed therapy, most particularly [36]. 

Conjunctival melanoma is very rare, comprising only 2% of ocular melanomas. Conjunctival melanomas are noted for lymphatic spread, distant hematogenous metastases, as well as invasion of the globe and orbital structures [37]. Prognosis is somewhat better than with uveal melanoma, with an overall survival of approximately 75%. Treatment is by early surgical excision, with cryotherapy of residual tissues [38]. As for uveal melanoma, adjuvant chemotherapy is not recommended, and systemic metastatic disease does not respond to cytotoxic chemotherapy. T-cell-directed therapy has met with some success [36]. The value of FDG PET-CT in the management of conjunctival melanoma is not established, but several case reports suggest that metastatic lesions are hypermetabolic [39,40,41].

#### 2.3.3. Mucosal Melanoma

Mucosal melanomas constitute less than 2% of all melanomas [42]. The most frequent sites of mucosal melanoma include anorectal, vulvovaginal and sinonasal locations. However, mucosal melanoma can occur virtually anywhere in the gastrointestinal (GI), genitourinary (GU) or upper aerodigestive tracts (Figure 7, Figure 8 and Figure 9). Mucosal melanomas have a generally poor prognosis, with 95% presenting with or subsequently developing metastases and often responding poorly to treatment (Figure 10) [42]. The liver is a common site for metastatic disease, but the pattern depends somewhat on the primary site of melanoma. Most imaging strategies for mucosal melanoma rely on conventional imaging [43]. Available evidence supports that both the primary mucosal lesions as well as metastases are intensely hypermetabolic, similar to cutaneous melanoma [44,45]. Some reports of single institutional experience have advocated the use of FDG PET-CT in staging esophageal melanoma and anorectal melanoma [46,47,48]. With anorectal melanoma, there is support for using FDG PET-CT for patient selection prior to extensive curative-intent abdominal perineal resection [49]. There may also be some value in utilizing PET-CT to identify the source of a primary occult melanoma, which is often mucosal, in patients who present with nodal or metastatic disease.

### 2.4. Merkel Cell Carcinoma

Merkel cell carcinoma is an aggressive primary neuroendocrine skin cancer. Presenting as a painless mass under the skin, these often evade early diagnosis. Merkel cell carcinomas are rare, but most occur in in the elderly in the periorbital region in areas of solar skin damage. Treatment is by wide excision and adjuvant radiation. Nodal dissection is performed when lymph node involvement is present and chemotherapy when distant metastases are found [50]. Although short-term regression of metastatic disease may be accomplished by chemotherapy, the disease ultimately progresses. With distant metastatic disease, Merkel cell carcinoma is essentially uniformly fatal. 

Sites of involvement with Merkel cell carcinoma are typically strongly hypermetabolic on FDG PET-CT. EARLY metastatic involvement is common (Figure 11 and Figure 12). There are data to support the value of FDG PET-CT in the early detection of recurrence and in surveillance for stage III Merkel cell carcinoma [51]. An additional report communicated that FDG PET-CT altered stage in approximately 39% of cases and altered management in approximately 55% [52]. Immune checkpoint inhibitors have shown promise in clinical trials for Merkel cell carcinoma and FDG PET-CT has been used successfully as an imaging marker of response [53]. Somatostatin receptor (SSTR)-targeting PET tracers, such as [^68^Ga]Ga-DOTA-TATE, have been reported to be effective in imaging Merkel cell carcinoma, with performances similar to that of FDG PET [54].

### 2.5. Cutaneous Squamous Cell Carcinoma

Cutaneous squamous cell carcinoma (SCC) is the second most common dermal malignancy (after basal cell carcinoma) and is linked to cumulative sun exposure and immunosuppressive drugs. Actinic keratoses are considered potential premalignant lesions. The treatment of cutaneous SCC is by surgical excision and Mohs surgery. When treated early, overall 5-year survival is >90%. However, when lymph nodes are involved, 5-year survival falls to 25–40%. Prognosis is affected by the size and thickness of the primary tumor, desmoplastic growth and immunosuppression [55].

Cutaneous SCC and its metastatic lesions are typically strongly hypermetabolic on FDG PET-CT (Figure 13). The value of FDG PET-CT in cutaneous SCC is best established in the initial assessment of higher-risk disease. In a series of patients with cutaneous SCC of the head and neck with biopsy-proven confirmation, the sensitivity and positive predictive value (PPV) of FDG PET-CT was 100% and 100% for the primary lesion and 100% and 25% for nodal disease [56]. Inflammatory or reactive nodes are sources of potential false positives in the setting of non-cutaneous squamous cell carcinoma of the head and neck [57]. Transplant patients or others who are immunocompromised are at increased risk for multiple cutaneous SCCs (Figure 14). FDG PET-CT has been reported to not significantly change management in patients with head and neck cutaneous SCC known to have nodal disease [58]. Despite this report, the NCCN endorses the use of FDG PET-CT in the initial evaluation for nodal basin assessment for radiation treatment planning and to rule out distant metastases [59]. The NCCN makes no recommendation for the use of FDG PET-CT for surveillance, assessment of response to treatment or evaluation of recurrent disease. However, FDG PET-CT has been shown to result in a change in management in 28% of patients with recurrent cutaneous SCC [60].

## 3. Sarcomas

Sarcomas represent about 2% of malignancies in adults and 10–15% of malignancies in children and adolescents/young adults. They represent a heterogeneous group of tumors that arise from mesenchymal cells of various tissues. There are two major categories of sarcomas—soft tissue and bone. The WHO classification of sarcomas revised in 2020 includes over 80 adult and pediatric types of sarcoma and sarcoma-like tumors in these categories [61]. A complete discussion of all possible types is beyond the scope of this review. Magnetic resonance imaging (MRI) and computed tomography (CT) are typically the imaging modalities of choice in sarcomas. However, there are a number of principles regarding FDG PET-CT imaging in sarcomas that are broadly applicable to most sarcomas. Since metastatic disease may occur widely, whole-body (head-to-toe) imaging is recommended. Most soft tissue and bone sarcomas are metabolically active on FDG PET-CT and can be intense so in more aggressive sarcomas [62]. Patients have been shown to be upstaged by PET-CT, with identification of M1 disease in approximately 12%, compared to conventional imaging. However, even in sarcomas that are moderately aggressive, there is a wide range of metabolic activity which may limit the use of FDG PET-CT for sarcomas as a broad category. With dedifferentiation or malignant degeneration of a number of benign soft tissue neoplasms or low-grade sarcomas, metabolic activity typically increases. While it does not substitute for biopsy and histologic analysis, FDG PET-CT may be useful in directing biopsy to the most metabolically active region and in suggesting transformation. Gastrointestinal stromal tumor (GIST, which is considered a sarcoma) and sarcomas of the uterus are addressed separately in the third and fourth articles of this review series.

Most of the medical evidence regarding the value of PET imaging in sarcomas is related to the use of FDG PET-CT. However, other PET radiopharmaceuticals have also been noted to be of potential benefit in imaging sarcomas. For example, both GIST tumors and pleomorphic sarcomas have been reported to show significant uptake of prostate-specific membrane antigen (PSMA)-binding ligands on PET-CT [63,64]. [^18^F]F-NaF PET-CT is extremely sensitive for areas of bone involvement in sarcomas. It has been proposed that a dual-tracer approach (simultaneous injection) of NaF and FDG PET may provide advantages when defining both soft tissue and bone sites of sarcoma involvement [65].

### 3.1. Soft Tissue Sarcomas

Soft tissue sarcomas account for approximately 1% of adult malignancies. There are more than 50 types of aggressive soft tissue sarcomas that are capable of metastatic spread. In addition, there are many intermediate sarcomas that may be locally invasive but which do not metastasize. Finally, a wide variety of benign soft tissue tumors also exist, some of which may undergo malignant degeneration into sarcomas. The most common soft tissue sarcomas are fibrosarcoma, typically a pediatric tumor, and malignant fibrous histiocytoma (now classified as undifferentiated pleomorphic sarcoma) originating from fibrous tissue. Liposarcoma originates from adipose tissue, leiomyosarcoma from smooth muscle cells and rhabdomyosarcoma from skeletal muscle. Hemangiopericytoma (solitary fibrous tumor) originates from perivascular cells. Sarcomas arising from blood vessel elements are Kaposi’s sarcoma, angiosarcoma and lymphangiosarcoma. Synovial sarcomas arise from synovial tissues. Nerve sheath tumors can be benign or undergo malignant transformation. Each of these tumors has characteristic biological, molecular, histologic, clinical and imaging features. 

In general, MRI is the modality of choice in characterizing most soft tissue sarcomas, although FDG PET-CT is typically intensely hypermetabolic in high-grade soft tissue sarcomas. The NCCN specifically endorses the use of FDG PET-CT in certain aspects of soft tissue sarcoma management [66]. For example, the NCCN also acknowledges that FDG PET-CT may be useful in staging and assessing the response to treatment of soft tissue sarcomas. FDG PET-CT has been shown to be effective in assessing the response to systemic therapy of soft tissue sarcomas. It has been reported that a baseline maximum standardized uptake value (SUVmax) of ≥6.0 or a decline of <40% are associated with high risk for recurrent soft tissue sarcoma treated with neoadjuvant chemotherapy [67]. After one cycle of chemotherapy, a reduction in SUVmax of <35% is predictive of a higher risk (33%) of poor histopathological response [68]. Interim (during treatment) FDG PET-CT may therefore allow for an earlier adjustment in therapeutic approach. 

#### 3.1.1. Undifferentiated Pleomorphic Sarcoma

Previously known as malignant fibrous histiocytomas, undifferentiated pleomorphic sarcomas typically arise in older adults, with a male predominance. They can arise at sites of prior radiation treatment. There is some histologic variability. Most of the tumors are intramuscular and can occur widely, though most often they occur in the extremities and torso. They can be associated with periodic hypoglycemia and can show rapid growth during pregnancy. Local imaging is typically performed by MRI, with and without contrast. The NCCN supports the use of FDG PET-CT in staging to evaluate for distant metastases (Figure 15) [66]. Treatment is typically performed by wide en bloc surgical resection, with limb-sparing if possible, followed by radiation. Combined modality chemotherapy or adjuvant chemotherapy is not typically utilized, except in the context of clinical trials. Resection or percutaneous ablation is frequently utilized for pulmonary metastases. High FDG uptake in undifferentiated pleomorphic sarcomas has been reported, and FDG PET is broadly endorsed for staging of soft tissue sarcomas in general [62,66,69,70]. However, the value of FDG PET-CT in specific series of undifferentiated pleomorphic sarcomas has not been addressed.

#### 3.1.2. Leiomyosarcoma

Leiomyosarcomas arise from smooth muscle and are non-gastrointestinal mesenchymal stromal tumors. These tumors account for 7% of soft tissue sarcomas, with an average survival of 5 years [71]. Non-uterine leiomyosarcomas can occur widely, including throughout the gut, involving vascular structures, but most often in the pelvis and retroperitoneum. Surgery is usually the treatment of choice, if possible. Radiation therapy and chemotherapy can be employed but are poorly effective. Long-term monitoring after treatment is typically required, even with low-grade leiomyosarcomas, which can be stable for years and then progress with metastases (Figure 16). FDG PET-CT may play a role in the management of leiomyosarcoma. There is evidence that higher degrees of metabolic activity are associated with higher-grade and larger tumors and with worse prognoses [72,73,74]. The role of FDG PET-CT in the surveillance and therapeutic assessment of leiomyosarcomas has not yet been established.

#### 3.1.3. Angiosarcoma

Sarcomas arise from vascular elements. Angiosarcomas include primary angiosarcomas of the aorta, great vessels and heart and are typically aggressive. Epithelioid and lymphangiosarcomas are slower-growing and less aggressive vascular sarcomas. In addition to the heart and large vessels, angiosarcomas can arise in many other organs, including the skin, lungs, bones, breasts, liver and spleen (Figure 17). Aggressive angiosarcomas can be intensely hypermetabolic on FDG PET-CT. On FDG PET-CT, they can mimic lymphoma and metastatic disease from other tumors. Angiosarcomas of the heart or great vessels often metastasize by hematogenous spread, most particularly to vascular beds downstream from the site of origin (Figure 18). However, some of these tumors are more well-differentiated and can exhibit lower metabolic activity [74]. FDG PET-CT has been reported to show lower activity with cardiac lymphoma than with cardiac angiosarcoma [75]. However, biopsy would likely be required to differentiate between the two entities. Although no systemic studies have been performed to evaluate the role of FDG PET-CT in angiosarcomas and related tumors, there are multiple reports of prominent metabolic activity being associated with angiosarcomas that have contributed to the management of patients in specific case reports and small series [76,77].

#### 3.1.4. Kaposi Sarcoma

Kaposi sarcoma (KS) is an angio-proliferative spindle cell tumor derived from immune and endothelial cells infected with human herpes virus type 8 (HHV-8). It occurs in conjunction with human immunodeficiency virus (HIV) infection and in immune-compromised patients. Sporadic (classical) cases also occur which are more indolent and typically confined to the skin. The disease is endemic in parts of Africa. The treatment for acquired immunodeficiency syndrome (AIDS)-related KS, which can be aggressive and widespread, is highly active antiretroviral therapy (HAART), sometimes in combination with chemotherapy. There is evidence that FDG PET-CT can be used to monitor the effectiveness of treatment and human herpes virus (HHV)-8 viral load [78,79]. In addition to KS, multicentric Castleman’s disease (also associated with HHV-8) can occur in HIV patients and it may be difficult to distinguish between these two entities via FDG-PET. In addition to KS, other complications of AIDS include benign lymphoproliferative disorder, lymphoma and EBV-associated lymphadenopathy. All may appear hypermetabolic on FDG PET-CT and coexist in various combinations [80]. KS typically appears as hypermetabolic dermal nodules and plaques on FDG PET-CT but is often associated with adenopathy and may display numerous system sites of involvement or metastatic disease (Figure 19).

#### 3.1.5. Rhabdomyosarcoma

Rhabdomyosarcoma arises from primitive mesenchymal tissue capable of differentiating into skeletal muscle. It is the most common soft tissue sarcoma. Ninety percent of cases occur in children and young adults. In older patients, rhabdomyosarcomas tend to be more aggressive. Treatment usually involves initial surgery and radiation, depending upon the extent of disease and the site of involvement. Chemotherapy may be utilized as well in high-risk patients. Metastatectomy or percutaneous ablative techniques may be performed for isolated pulmonary metastases. Sinonasal rhabdomyosarcoma is most common in children. Sinonasal plasmacytoid rhabdomyosarcoma in adults is in the category of small blue cell tumors and, although rare, can masquerade as other sinonasal tumors (Figure 20) [81]. Rhabdomyosarcomas tend to be moderate in terms of metabolic activity on FDG PET-CT. FDG PET-CT has been shown to be effective as well as more accurate than conventional imaging (bone scan, radiographs, CT, MRI) in staging and restaging rhabdomyosarcoma and this is acknowledged by the NCCN [82,83]. FDG PET-CT has been shown to be useful in monitoring response to therapy for pediatric rhabdomyosarcomas, although it has not been as well studied in adults [84]. However, the magnitude of metabolic activity pre-treatment does not predict outcome in rhabdomyosarcoma in children [85].

#### 3.1.6. Liposarcoma

Liposarcoma is the most common soft tissue sarcoma in adults. Most cases arise de novo, although a small number may arise from preexisting lipomas. Liposarcoma exists as a well-differentiated subtype (occurring in soft tissues of both the limbs or retroperitoneum), a myxoid and/or round cell subtype (occurring predominantly in the limbs), and a pleomorphic, dedifferentiated subtype (primarily in the retroperitoneum). The latter is aggressive, with a poorer prognosis. Hibernomas, which are benign lipomas with brown fat characteristics, are often FDG-avid and found incidentally, but a distinction between these and low-grade liposarcomas cannot be made by FDG PET-CT (Figure 21). In general, metabolically active lipomas should be regarded as potentially malignant and biopsied if large, growing or symptomatic, and otherwise followed for stability. It has been suggested that there are significant metabolic differences between low- and high-grade liposarcomas on FDG PET-CT. However, other data suggest that there is significant overlap between benign and malignant lipomatous tumor subtypes (Figure 22, Figure 23 and Figure 24) [86]. Although the NCCN recommends consideration of the use of FDG PET-CT to help distinguish well-differentiated from dedifferentiated liposarcoma, the best use of FDG PET-CT is to help guide biopsy to the most metabolically active portion of the tumor [86]. Treatment of liposarcoma is typically by surgery and radiation. Chemotherapy may be of some benefit in patients with metastatic dedifferentiated liposarcoma [87]. The role of FDG PET-CT in assessing treatment response and in the surveillance of liposarcoma has not been established.

#### 3.1.7. Synovial Sarcoma

Synovial sarcomas are rare soft tissue tumors. Advanced-stage synovial cell sarcomas have a poor prognosis, but lower-stage disease has a 70% 5-year survival. Over 90% of cases are associated with a fusion of the SYT/SSX genes t(x;18)(p11,q11). Synovial cell sarcomas occur as spindle cell fibrous and glandular/epithelial forms. Synovial cell sarcomas tend to occur around the large joints, such as the knees, but also in other areas, including the hands, feet, neck, pericardium and pleura (Figure 25 and Figure 26). Treatment consists of aggressive surgery, radiation and, in tumors >5 cm, chemotherapy. Higher metabolic activity on FDG PET-CT (SUVmax > 4.4) has been associated with a worse prognosis [88,89]. This could affect the aggressiveness with which a multimodality therapy approach is pursued. Beyond this, the role of FDG PET-CT in synovial sarcomas has not been well studied. 

#### 3.1.8. Nerve Sheath Tumors

Benign nerve sheath tumors, including neurofibromas and schwannomas, can occur sporadically. Neurofibromatosis type-1 (NF1), which is caused by a deletion on chromosome 17q, is associated with multiple neurofibromas. Schwannomatosis is seen in NF2. Schwannomas are benign, as are most neurofibromas. Gangioneuromas arise from the sympathetic chain and are typically benign (Figure 27). However, neurofibromas can undergo malignant degeneration, often heralded by pain and increasing mass or nerve symptoms. Malignant nerve sheath tumors typically occur in adults and can be induced by prior radiation [90]. Fifty percent of these tumors occur in NIF-1 patients, who have a risk of 8–13% of developing malignant peripheral nerve sheath tumors [91]. A challenge for FDG PET-CT is to differentiate benign from malignant nerve sheath tumors. In one report, all lesions with an SUVmax <4.4 were benign, while those with an SUVmax >8.1 were malignant. Lesions between the two cut-off values were a mixture of benign and malignant nerve sheath tumors (Figure 28 and Figure 29) [92]. However, these cut-off values have not been prospectively and independently validated and therefore probably should not be strongly relied upon at this time. Other imaging variables that distinguish benign from malignant nerve sheath tumors should also be considered when evaluating PET-CT images, including the size of the lesion. Malignant nerve sheath tumors are typically >5 cm in diameter, have indistinct margins and are not usually associated with a specific identifiable nerve root (which is a factor typically seen with benign nerve sheath tumors) [93]. Correlative imaging features must also be considered, such as those shown by MRI. No differences in signal intensity on MRI can distinguish benign vs. malignant tumors, but MRI findings for malignant nerve sheath tumors are often associated with changes in adjacent soft tissues. Some nerve sheath tumors are surgically removed, even if benign, due to local compressive symptoms. For example, dumbbell neurofibromas of the spine (typically the thoracic spine) have both paraspinous and intraspinous components that can lead to nerve root or spinal cord compression (Figure 30). 

### 3.2. Bone Sarcomas

The most common bone sarcomas are chondrosarcoma, osteosarcoma and Ewing sarcoma. All can metastasize to bone and to soft tissues. 

#### 3.2.1. Chondrosarcoma

Chondrosarcoma is an inclusive term for a variety of cartilage-containing tumors. They can be broadly classified as low- (grade I), intermediate- (grade II), or high-grade (grade III) or dedifferentiated (grade IV) tumors. Chondrosarcomas can arise de novo or by malignant degeneration of enchondromas or cartilage-capped exostoses. Malignant degeneration is often heralded by pain in the region. MRI may be utilized for evaluation. Intermediate and higher grade enchondromas are also painful and present as a large, expansile mass, sometimes with an accompanying soft tissue component. Myxoid chondrosarcomas arise from soft tissues rather than bone (Figure 31). Biopsy can be performed but due to heterogeneity within the tumor and the coexistence of benign cartilaginous elements may be inconclusive. Treatment of presumed low-grade chondrosarcomas is by curettage, with complete surgical resection of presumed higher-grade lesions, which may also be required for diagnosis [94]. Radiotherapy and chemotherapy may be used when the lesion is unresectable or there is metastatic disease. No chemotherapy guidelines are issued for grade I–III lesions. For grade IV, dedifferentiated tumors, NCCN chemotherapy guidelines for chondrosarcoma follow those for osteosarcoma. For mesenchymal histologic subtypes, Ewing sarcoma guidelines are observed [95]. 

Enchondromas may have no activity or may be mildly metabolically active on FDG PET-CT, making distinction between these and low-grade chondrosarcomas difficult (Figure 32). The NCCN endorses the use of FDG PET-CT in staging conventional, poorly differentiated or dedifferentiated chondrosarcoma, and suggests that a whole-body bone scan also be performed if the FDG PET-CT scan is negative [95]. The NCCN does not address the role of FDG PET-CT in evaluating response to treatment or in surveillance of chondrosarcomas. FDG PET-CT has been explored to differentiate benign from malignant chondroid lesions, as well as different grades of chondrosarcoma [96]. Based on this report, SUVmax >4.4 predicted higher-grade chondrosarcoma with a sensitivity of 99%. SUVmax for benign enchondroma was 1.6 (+/− 0.7) compared to an SUVmax of 4.4 (+/− 2.5) for chondrosarcoma. The usefulness of magnitude of metabolic activity in distinguishing between different grades of chondrosarcoma has also been addressed. Although the magnitude of uptake on PET CT is significantly different for benign versus high-grade lesions, there is some overlap between categories [96]. Perhaps the most significant report that supported the use of PET-CT in chondrosarcoma showed that the SUVmax of the primary lesion, using an SUVmax cut-off of 4.0, could predict the likelihood of relapse with a sensitivity, specificity, positive predictive value (PPV) and negative predictive value (NPV) of 90%, 76%, 64% and 94%, respectively. When SUVmax was combined with histopathological tumor grade, these values were further improved to 90%, 95%, 90% and 95% [97]. This finding is significant in that it may pave the way for the selection of high-risk patients who may benefit most from clinical trials or aggressive combined modality treatment. 

#### 3.2.2. Osteosarcoma

Osteosarcoma is a potentially deadly form of bone cancer that typically arises in the distal femur, proximal tibia and proximal humerus of adolescents, and less commonly in the skull, jaw and pelvis. There are multiple variants of osteosarcoma, including conventional types and telangiectatic, multifocal, parosteal and periosteal subtypes. Most cases of osteosarcoma are without a known cause, although patients with certain conditions are at higher risk, including bone dysplasias such as Paget’s disease, fibrous dysplasia, enchondromatoses, multiple hereditary exostoses and the germ-line form of retinoblastoma. Five-year survival depends upon age—younger age being more favorable—with a value of 61.1% for age <25 years and a severe decline in disease-specific survival for elderly patients [98]. Diagnosis is made by biopsy. Staging typically utilizes MRI for identification of “skip” lesions and soft tissue involvement, CT for assessment of visceral metastases and radionuclide bisphosphonate bone scan for identification of osseous metastases or multifocal disease. Although FDG PET-CT is effective in staging osteosarcoma, it is reported that there are no significant differences between FDG PET-CT, whole-body MRI and whole-body bisphosphate bone scan in detecting skeletal metastases in osteosarcoma [99]. Nonetheless, the NCCN endorses the use of FDG PET-CT for staging osteosarcoma (Figure 33 and Figure 34) [95].

Guidelines for treatment of osteosarcoma are published by the NCCN and the European Society of Medical Oncology (ESMO) [95,100]. Treatment of osteosarcoma involves surgical resection with bone reconstruction. Amputation is typically reserved for recurrence. Neoadjuvant chemotherapy is typically given prior to resection to reduce the size of the primary tumor. Adjuvant chemotherapy may be given post-operatively if there has been a good response to preoperative chemotherapy [101]. However, most clinical trials have focused on children and the optimal therapeutic approach in adults is in evolution. In predicting response to chemotherapy, a post-treatment SUVmax of ≤1.5 or a post treatment decrease in SUVmax of ≥50% relative to the pre-treatment value predicts a histologic response to chemotherapy [102]. The NCCN also supports the use of FDG PET-CT for monitoring the response of osteosarcoma to treatment [95]. 

#### 3.2.3. Ewing Sarcoma

The primary bone tumor, Ewing sarcoma, its soft tissue counterparts, peripheral primitive neuroendocrine tumor (PNET), and malignant small cell tumors of the thoracopulmonary region (Askin tumor) represent a related group of tumors (Ewing sarcoma family) with similar immunohistochemistry, genetic features and cellular origin. The bone tumors arise spontaneously with pain and swelling, typically involving the flat bones and the diaphyseal region of larger long bones. The age of occurrence of most tumors is the late teens and 20s. Most patients present with localized disease, which carries an overall survival rate of 60%. Patients with metastatic disease have an overall survival of <25%. Due to a high relapse rate with local treatment, such as surgery and radiation, systemic chemotherapy is typically utilized. Treatment guidelines have been published by NCCN and ESMO and are frequently revised with new medical evidence to support the revisions [95,100]. The imaging evaluation of Ewing sarcoma is typically multimodal and includes MRI imaging of the primary site, chest CT and, as endorsed by the NCCN, FDG PET-CT for initial staging, with a recommendation for head-to-toe imaging (Figure 35) [95]. It has been shown that FDG PET-CT, without bone marrow biopsy, has a sensitivity of 100% and a specificity of 96% for the identification of sites of disease [103]. FDG PET-CT has been successfully used to evaluate response of Ewing sarcoma to treatment in a number of clinical trials [104]. Progression-free survival following neoadjuvant and adjuvant chemotherapy was 80% for patients with a post-treatment SUVmax of <2.5 and 33% for those with an SUVmax of ≥2.5 [105].

## 4. Neuroendocrine Tumors

### 4.1. Epidemiology and Agents

Neuroendocrine neoplasms or tumors (NETs) are a group of heterogeneous tumors that arise from the disseminated endocrine cell system, primarily from gastroenteropancreatic (GEP) organs. These tumors often produce and secrete biologically active peptides and amines which can induce clinical syndromes. Alternatively, they can be functionally inactive. The incidence and prevalence of NETs is rising in the United States. Based upon the National Cancer Institute’s Surveillance, Epidemiology, and End Results (SEER) Program data, the age-adjusted incidence rate has increased 6.4-fold from 1973 (1.09 per 100,000) to 2012 (6.98 per 100,000) and the age-adjusted incidence of NETs was 1.09 per 100,000 persons in 1973 and had increased to 6.98 per 100,000 persons by 2012. The precise etiology for this rise is not entirely clear but may in part be due to the increased diagnosis of early-stage disease and stage migration [106]. Survival for all NETs has also improved with therapeutic innovations, particularly for advanced-stage gastrointestinal and pancreatic NETs. New developments in diagnostic capabilities, including somatostatin receptor (SSTR) PET-CT play an important role in improved patient management. 

Nuclear medicine is playing an increasingly essential role in the management of NETs through improved diagnostic and therapeutic capabilities. Novel nuclear imaging and therapeutic techniques have targeted peptide hormone receptors which are overexpressed in selected tumor types, including NETs, and have therefore enabled in vivo tumor targeting for diagnostic imaging and therapies. The majority of well-differentiated NETs overexpress somatostatin receptors (SSTR) on their surface and are hypervascular. The overexpression of SSTRs on NETs has enabled improved diagnostic imaging capabilities with peptide hormone somatostatin analogue-binding (SSA)-based imaging and therapy (e.g., Lu-177 peptide receptor-based radionuclide therapy, such as ^1^[^77^Lu]Lu-DOTA-TATE PRRT). For decades, SSTR imaging was performed with [^111^In]In-pentetreotide (OctreoScan^®^, Mallinkrodt Pharmaceuticals, Staines, UK) [106,107]. This approach has been largely supplanted by two PET radiopharmaceuticals that are FDA-approved in the US for imaging well-differentiated neuroendocrine tumors. Both bind to the SSTR (primarily SSTR subtype 2). These are [^68^Ga]Ga-DOTA-TATE (NETSPOT^®^, Novaris Pharmaceuticals Corp, East Hanover, NJ, USA), and [^64^Cu]Cu-DOTA-TATE (Detectnet^®^, Curium Pharma, Paris, France) [108]. SSTR PET-CT offers significant advantages in terms of sensitivity and resolution of NET tumors when compared with the single photon emission computed tomography (SPECT) agent, OctreoScan^®^ (Figure 36). 

Gallium-68 is a generator-produced positron emitter with a short half-life of 68 min compared to 2.8 days for In-111. This enables on-site, as-needed availability of the radiotracer and a lower radiation dose for the patient compared to In-111, with an effective dose of 2.1 mSv for a 100 MBq administration [109,110]. Ga-68 is chelated to a variety of somatostatin analogs by means of the organic compound 1,4,7,10-tet- raazacyclododecane-1,4,7,10-tetraacetic acid (DOTA), resulting in higher-affinity radiopharmaceuticals that are typically abbreviated as DOTA-TATE (GaTate), DOTA-TOC (GaToc) and DOTA-NOC (GaNoc) [111]. DOTA-TATE is the construct that is FDA-approved in the US and can be conjugated either with Ga-68 or Cu-64. DOTA-TATE has the highest affinity for SSTR subtype 2 (SSTR 2), which tends to have the greatest overexpression in NETs [112]. The other SSTR-binding constructs are used primarily outside the US and in research studies in imaging NETs. 

[^64^Cu]Cu-DOTA-TATE is a new somatostatin receptor-based radiotracer that was approved by the FDA on 8 September 2020 for the localization of somatostatin receptor-positive neuroendocrine tumors (NETs) in adult patients [113]. Cu-64 has a shorter positron range than Ga-68, which theoretically leads to a better spatial resolution. The increased physical half-life of 12.7 h for Cu-64 compared to 68 min for Ga-68 makes [^64^Cu]Cu-DOTA-TATE attractive for widespread clinical adoption as it can be produced centrally and shipped to sites throughout the US. In a prospective study by Johnbeck et al., where [^64^Cu]Cu-DOTA-TATE and [^68^Ga]Ga-DOTA-TOC PET/CT were compared on a head-to-head basis, [^64^Cu]Cu-DOTA-TATE demonstrated the same per patient sensitivity as [^68^Ga]Ga-DOTA-TOC in detecting NETs, but significantly more lesions were detected with [^64^Cu]Cu-DOTA-TATE [114]. A longer shelf-life of more than 24 h for [^64^Cu]Cu-DOTA-TATE compared to [^68^Ga]Ga-DOTA-TATE and a scanning window of at least 3 h for [^64^Cu]Cu-DOTA-TATE has led to its increased adoption in multiple institutions. Beyond the physical differences that contribute to the detectability of lesions, the biodistributions of both DOTA-TATE constructs are essentially the same. We shall refer to PET-CT scans performed with these collective constructs as SSTR PET-CT scans.

### 4.2. Normal Physiologic and Benign Patterns

In order to adequately assess the presence of NETs as well as the extent of the patient’s disease burden with SSTR PET-CT imaging, it is important to understand the normal physiologic and benign patterns of uptake for SSA-based PET-CT. The most intense benign SSTR PET-CT uptake is present in the spleen, followed by the adrenal glands, kidneys and pituitary gland. Moderately intense uptake is present in the liver, salivary glands and thyroid gland, with variable degrees of gastric and small and large intestinal uptake. Prominent uptake is also seen in the uncinate process of the pancreas under normal conditions (Figure 37) [115]. This biodistribution of these agents is due to a combination of SSTR binding and physiologic factors, such as excretion. Uptake in the endocrine organs, salivary glands and spleen is mediated by SSTR2 expression, while uptake in the kidneys, liver and bowel is physiologic, with the exact mechanism causing some of this uptake remaining unclear. Additional benign patterns are also important to recognize to exclude interpretive errors and misdiagnosis of malignant disease. Commonly encountered benign foci of uptake include prominent pancreatic uncinate process activity, inflammatory processes, osteoblastic activity, including degenerative disease, trauma, vertebral hemangioma and splenunculi (splenules, accessory splenic tissue, splenosis) or meningiomas [112]. Some thyroid uptake is also normal.

### 4.3. Gastroenteropancreatic (GEP) Neuroendocrine Tumors

Gastroenteropancreatic (GEP) neuroendocrine tumors or GEP NETs are NETs arising from the gastrointestinal tract and pancreas. GEP NETs are a heterogeneous group of malignancies with many different subtypes ranging from indolent, well-differentiated NETs to aggressive, poorly differentiated neuroendocrine carcinomas (NECs). All GEP NETs can metastasize, and even indolent tumors can lead to significant morbidity due to the secretion of bioactive hormones [116]. NETs are classified according to cell morphology and proliferation index (Ki67) as grade 1 (low-grade, Ki67 ≤ 2%), grade 2 (intermediate-grade, Ki67 3–20%) and grade 3 (high-grade, Ki67 > 20%). Both grades 1 and 2 are classified as differentiated NETs, while grade 3 is considered poorly differentiated. Differentiated NETs (G1–2) have a more favorable behavior compared to poorly differentiated G3 and neuroendocrine carcinomas (small and large cell varieties) [117].

Multiple studies have demonstrated improved diagnostic capabilities of SSTR PET-CT compared to conventional imaging, including contrast-enhanced CT, in the diagnosis of GEP NETs (Figure 38). For example, Geijer et al. performed a meta-analysis of 22 SSTR PET-CT studies that included a total of 2,105 patients and showed sensitivity to be 93% and specificity 96%. They determined that SSTR PET-CT has better diagnostic performance for evaluation of NETs in the thorax and abdomen than SSTR scintigraphy [118]. MRI of the abdomen is a preferred imaging modality in evaluating hepatic burden of metastatic disease and may allow identification of tiny lesions that are inapparent with PET-CT. SSTR PET is generally quite similar to MRI in assessment of hepatic burden of disease but also allows whole-body imaging for possible metastatic disease (Figure 39). However, with MRI, the apparent size of lesions can vary with contrast timing and reproducibility in contrast administration is critical in following the size of hepatic lesions. 

Additionally, SSTR PET-CT may ultimately play a role in restaging and assessing tumor characteristics in vivo. Studies have demonstrated excellent correlation between SUV values for SSTR PET-CT and SSTR2 gene expression, suggesting that SUV quantification may help assess tumor characteristics and aid treatment decisions [119,120,121,122]. A number of studies have shown that the comparison of SSTR and FDG PET-CT systemically aided assessment of tumor grades and potential sites of higher-grade transformation [123,124]. This highlights the potential role multi-tracer PET-CT can play in better assessing the heterogeneous character of a patient’s disease as well as the limitations of the grading standard based upon a biopsy at a single site. Dual-tracer imaging with SSTR PET ligands and FDG PET has been shown to be useful in predicting the aggressive nature and resectability of pancreatic neuroendocrine tumors in patients undergoing pre-operative assessment. In those instances, FDG PET is useful in identifying more aggressive features and angioinvasion and SSTR PET-CT shows superior detection of metastasis in patients considered for resection [125,126,127,128]. Dual-tracer FDG and SSTR PET-CT may also be useful in the appropriate selection of patients for PRRT therapy (Lutathera^®^). Typically, activity in the tumor that is greater than that in the liver is recommended in selection of patients who may best benefit by PRRT. SSTR PET-CT is excellent for well-differentiated NET imaging, with typically shows intense activity. Well-differentiated NETs characteristically are low in metabolic activity on FDG PET-CT (Figure 40). These patients would be regarded as potentially good candidates for ^177^Lu DOTATATE (Lutathera^®^). Poorly differentiated NETs are typically low in activity on SSTR-PET and are better imaged with FDG PET-CT (Figure 41). In these patients, it could be predicted that Lutathera^®^ treatment would not be effective. 

### 4.4. Bronchial Carcinoid

Bronchopulmonary NETs or pulmonary carcinoids (PCs) are less common than GEP NETs, accounting for approximately 1–2% of all lung malignancies and 20–25% of NETs [106,129]. Pulmonary carcinoids (PCs) are histologically classified into well-differentiated typical carcinoids (TCs) and poorly differentiated atypical carcinoids (ACs). The distinction TCs and ACs could potentially significantly impact treatment planning and prognosis, with ACs being associated with poorer prognoses [130]. PET-CT is playing an increasing role in the evaluation and workup of PCs, with FDG PET-CT typically demonstrating greater uptake in poorly differentiated lung NETs (ACs), while SSTR PET-CT typically demonstrates greater uptake in TCs [131,132].

Kayani et al. reviewed the imaging findings of 18 consecutive patients with pulmonary NETs (11 typical carcinoids, 2 atypical carcinoids, 1 large cell neuroendocrine tumor, 1 small cell neuroendocrine carcinoma, 1 non-small cell lung cancer with neuroendocrine differentiation and 2 cases of diffuse idiopathic pulmonary neuroendocrine cell hyperplasia) who underwent SSTR as well as FDG PET-CT scans [129]. They found that SSTR PET-CT was superior to FDG PET-CT in typical histologically well-differentiated bronchial carcinoids and that it could delineate an endobronchial tumor from an adjacent atelectasis and secondary obstructive changes (Figure 42). A good correlation was also observed between the grades of pulmonary neuroendocrine tumors with high uptake on SSTR PET-CT in all typical carcinoids and variable and often low uptake on FDG PET-CT. Some studies have suggested that the SUV max ratio between SSTR and FDG PET-CT may have utility in predicting the histologic subtype of PC tumors [133]. 

### 4.5. Pheochromocytoma and Paraganglioma

Neuroectodermal tumors, including pheochromocytomas and paragangliomas (PPGLs), have demonstrated high uptake on SSTR imaging for decades, initially based upon early experience with Octreoscan^®^ imaging [134]. [^131/123^I]I–(3 Iodobenzyl)-guanidine (MIBG) was more commonly used for pheochromocytomas. Now, SSTR PET-CT imaging is becoming the functional imaging modality of choice at many centers for both paragangliomas and larger adrenal pheochromocytomas.

In 2014, Sharma et al. evaluated the diagnostic accuracy of SSTR PET-CT in 62 patients with clinically suspected pheochromocytoma. They found that SSTR PET-CT had a sensitivity, specificity and accuracy of 92%, 85% and 90%, respectively. The lesion-based accuracy of SSTR PET-CT for pheochromocytoma was significantly higher than ^131^I-MIBG imaging (91.1 vs. 66.6%, *p* = 0.035). While [^123^I]I-MIBG, which has replaced [^131^I]I-MIBG, is still used, this study supported the improved diagnostic accuracy and image quality of SSTR PET-CT compared to MIBG SPECT-CT imaging (Figure 43) [135]. MIBG SPECT-CT remains important for the determination of candidates for MIBG therapy and for the evaluation of small adrenal lesions due to the high activity observed in normal adrenal glands on SSTR PET-CT. SSTR PET-CT is preferred for larger adrenal pheochromocytomas and paragangliomas. 

Clinical PRRT therapeutic trials targeting SSTR are in development for metastatic PPGLs, which will require confirmation of SSTR expression by SSTR PET-CT. A recent meta-analysis by Han et al. assessed the performance of SSTR PET in the detection of PPGLs in thirteen studies for qualitative synthesis. The team found that, with the exception of polycythemia/paraganglioma syndrome, where the detection rate of SSTR PET-CT was only 35%, in other PPGLs, per-lesion detection rates of SSTR PET were consistently higher (92–100%) than other imaging modalities, including 18F-fluorohydroxyphenylalanine ([^18^F]F-DOPA) PET, FDG PET-CT and ^123/131^I-MIBG scintigraphy. These findings support the utilization of SSTR PET/CT imaging as a first-line imaging modality for the primary staging or restaging of PPGLs with unknown genetic status [136,137]. Paragangliomas are variable with respect to the magnitude of metabolic activity shown on FDG PET-CT. Most show some degree of increased uptake on FDG PET-CT, although the magnitude of tumor activity is typically less than that seen with SSTR PET-CT (Figure 44). However, there is a significant cost differential between FDG and SSTR PET-CT, SSTR PET-CT being typically 2–3 times more expensive based solely on the price of the radiopharmaceutical. Therefore, FDG PET-CT is often preferred for the screening of paragangliomas in patients with SDHD mutations, which has been shown to be effective in most cases [138]. It should be mentioned that medullary thyroid cancer is discussed in the fifth article in this series. 

## 5. Conclusions

The sixth and final article in this series addresses the role of FDG PET-CT in non-lymphomatous dermal malignancies, in sarcomas and in neuroendocrine tumors. FDG PET-CT is critical in the staging, therapeutic assessment and surveillance of dermal malignancies. Since many sites of disease, particularly lymph nodes, can be tiny, PET-CT offers a clear advantage over CT alone in the assessment of metastases from dermal malignancies. FDG PET-CT also allows imaging of the entire body. This is important because dermal malignancies, particularly melanomas and Merkel cell carcinomas, can have unpredictable and broadly distributed patterns of metastatic spread, remote from the site of the primary lesion. As aggressive dermal malignancies, such as melanoma, are often treated with immune therapy/checkpoint inhibitors, it is important for the imaging provider to be aware of an ever-expanding body of literature regarding the adverse events that occur with these types of therapies and the effects of these immune therapies on FDG PET-CT scans, both with respect to the tumors themselves as well as the large number of systemic effects that can affect the results of FDG PET-CT scans. 

There is wide diversity in the types of sarcomas, with a number of them being quite rare (≤1 per 1,000,000) [139]. A discussion of all types is beyond the scope of this single article. This report focuses on those categories of sarcoma that are most commonly seen in clinical practice. Different types of sarcomas display disparate histologic and biological features, including degree of aggressiveness and magnitude of metabolic activity on FDG PET-CT. For this reason, it is necessary for the imaging practitioner to be cognizant of and up to date regarding the distinguishing imaging features, treatments, routes of spread and both expected and atypical imaging patterns for the specific type of sarcoma for which FDG PET-CT referral is made. It is also critical to have a clear understanding of the patient-specific clinical history, therapeutic events, results of other imaging studies, such as MRI, and typical or atypical histologic features of each patient’s tumor. All of these factors are important in providing accurate interpretations of FDG PET-CT scans for sarcomas.

In addition to FDG, a number of other radiopharmaceuticals play either a current or potential role in the staging, therapeutic assessment and surveillance of neuroendocrine tumors. The imaging signatures of neoplasms, sometimes derived from a multi-tracer approach, have potential for characterizing tumors from a biological perspective and in the selection of patients for appropriate therapies. Many other types of tumors display variable degrees of neuroendocrine features. It is anticipated that, with time and additional data, the role of traditionally neuroendocrine-specific imaging tracers may be expanded to include a greater diversity of tumors and possibly targeted therapies based on imaging signatures.

## Figures and Tables

**Figure 1 cancers-14-02835-f001:**
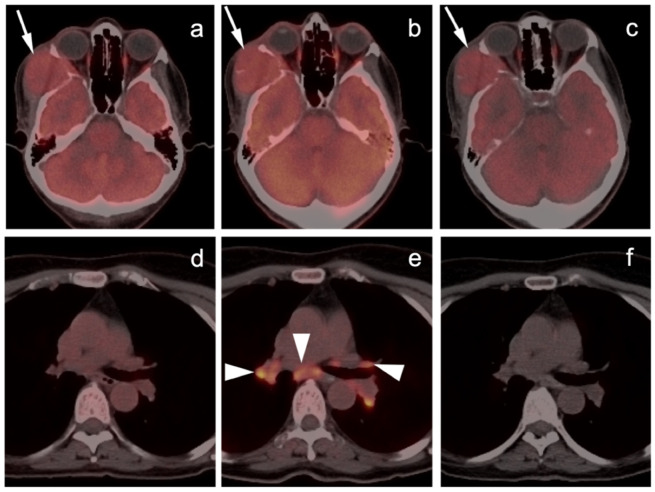
Sarcoidosis-like reaction to immunotherapy. ((**a**,**d**), left panels) Axial FDG PET-CT scan of the head (**a**) demonstrates a hypermetabolic mass (melanoma) invading the right temporalis muscle with lateral intra-orbital extension (white arrow). (**d**) No mediastinal adenopathy is present on axial FDG PET-CT of the upper chest. ((**b**,**e**), middle panels) Follow-up FDG PET scan while on immunotherapy. (**b**) Axial FDG PET-CT scan of the head demonstration that the melanoma in the right temporalis muscle is stable (white arrow). (**e**) Axial FDG PET-CT during immunotherapy demonstrates new hypermetabolic bilateral hilar and subcarinal adenopathy (white arrowheads), that was granulomatous on biopsy; ((**c**,**f**), right panels) Follow-up FDG PET-CT following discontinuation of immunotherapy shows stable melanoma ((**c**), white arrow) and resolution of hypermetabolic mediastinal and hilar adenopathy (**f**).

**Figure 2 cancers-14-02835-f002:**
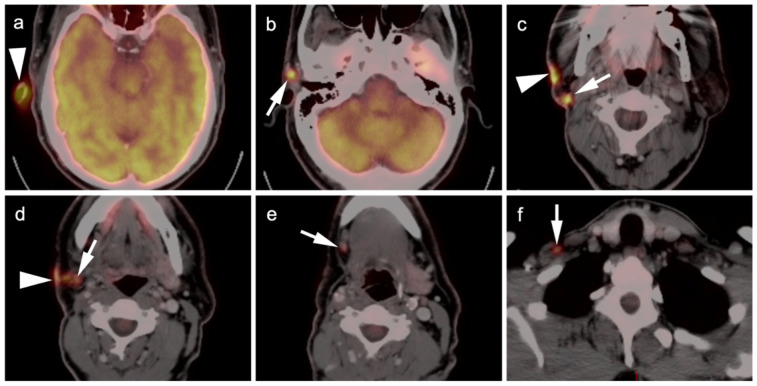
Melanoma of the right ear, metastatic to skin and multiple regional lymph nodes (patient had prior neck dissection for a previous melanoma). Six axial images from an FDG PET-CT scan of the head and neck are shown. (**a**) Primary tumor in the superior aspect of the right ear (white arrowhead). (**b**) Extension to a right preauricular lymph node (white arrow). (**c**) Extension along the skin (white arrowhead) with involvement of a parotid tail node (white arrow). (**d**) Further extension along the right skin (white arrowhead) with involvement of a right level 2 node. (**e**) Extension to a right submandibular node (white arrow). (**f**) Extension to a right supraclavicular node (white arrow).

**Figure 3 cancers-14-02835-f003:**
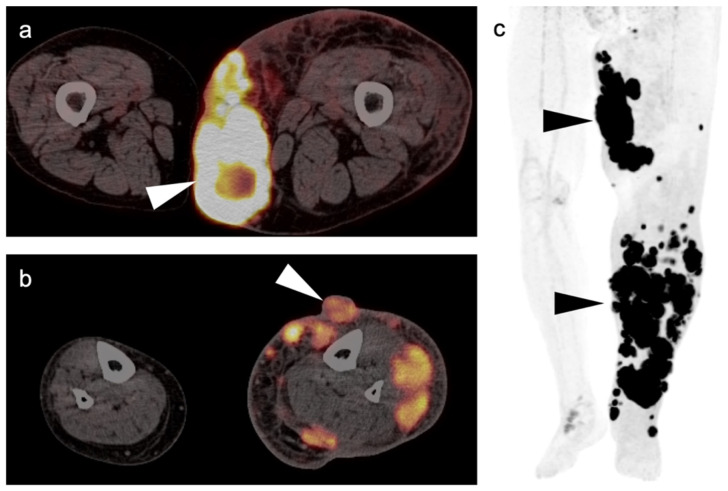
Melanoma of the left lower leg, metastatic to the lower extremity. (**a**) Axial FDG PET-CT images of the thighs show an intensely hypermetabolic tumor in the subcutaneous tissues of the left medial thigh (white arrowhead). (**b**) Axial FDG PET-CT image of the lower legs shows that the primary lesion is exophytic on the anterior left lower leg (white arrowhead), with multiple subcutaneous metastases, some extending into the muscles. (**c**) FDG PET maximum intensity projection (MIP) image demonstrates the intensity of metabolic activity throughout the tumor in the left lower extremity (black arrowheads).

**Figure 4 cancers-14-02835-f004:**
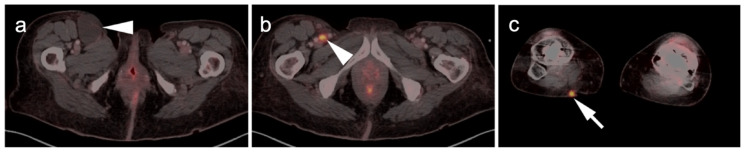
Identification of primary site of melanoma by FDG PET-CT. (**a**) Axial FDG PET-CT image of the low pelvis. A right inguinal mass was excised (melanoma on pathology), leaving a residual fluid collection (seroma or lymphocele, white arrowhead). (**b**) Axial FDG PET-CT image of the pelvis demonstrates a small residual hypermetabolic lymph node near the surgical bed (white arrowhead). (**c**) Axial FDG PET-CT image of the lower legs demonstrates the primary tumor (small and inconspicuous clinically) in the posterior upper right calf (white arrow).

**Figure 5 cancers-14-02835-f005:**
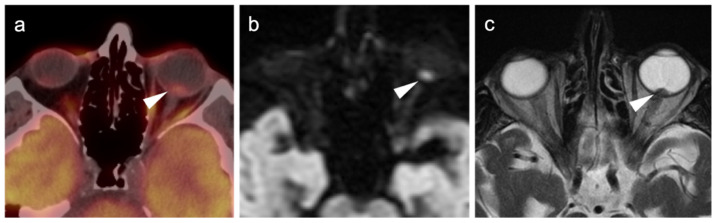
Retinal melanoma. (**a**) Axial FDG PET-CT image of the orbit shows the known left retinal melanoma to be only minimally hypermetabolic (white arrowhead). (**b**) Axial diffusion-weighted imaging (DWI) MRI of the orbit shows the retinal lesion to have high signal (diffusion restriction) (white arrowhead). (**c**) Axial fat-suppressed T2 (T2 FS) MRI image of the orbit shows a retinal nodule corresponding to the melanoma (white arrowhead).

**Figure 6 cancers-14-02835-f006:**
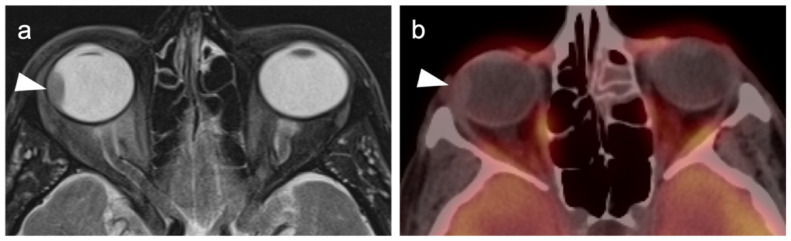
Choroidal melanoma. (**a**) An axial T2 MRI image of the orbits shows a small choroidal melanoma in the lateral inner aspect of the right globe (white arrowhead). (**b**) Axial FDG PET-CT image of the orbits shows no appreciable metabolic activity in the choroidal melanoma (white arrowhead).

**Figure 7 cancers-14-02835-f007:**
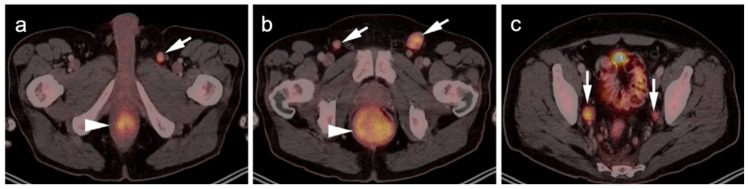
Anorectal melanoma. (**a**,**b**) Axial FDG PET-CT images of the pelvis. Metabolically active melanoma involving the rectum and anus (white arrowheads) with bilateral hypermetabolic inguinal nodes (white arrow). (**c**) Axial FDG PET-CT of the pelvis shows bilateral internal iliac hypermetabolic nodes (white arrows).

**Figure 8 cancers-14-02835-f008:**
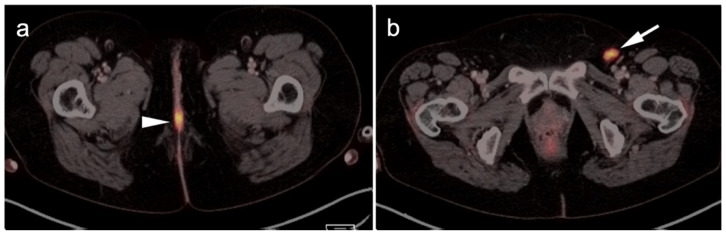
Vulvar melanoma. (**a**) Axial FDG PET-CT image of the pelvis shows a vulvar melanoma that is small but metabolically active (white arrowhead). (**b**) Axial FDG PET-CT of the pelvis also shows a hypermetabolic left inguinal node (white arrow), representing spread of disease.

**Figure 9 cancers-14-02835-f009:**
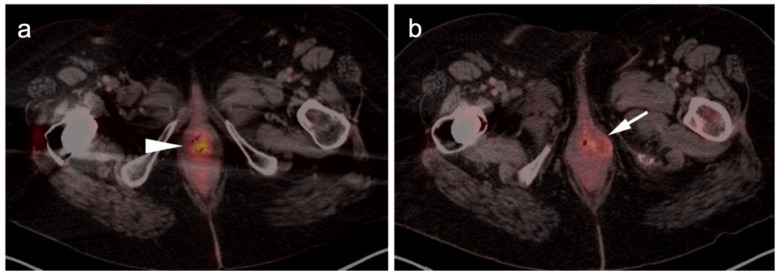
Vaginal melanoma. Axial FDG PET-CT images of the pelvis. (**a**) Vaginal melanoma at the introitus (white arrowhead) is mildly hypermetabolic. (**b**) One year following treatment, an intra-labial recurrence can be observed on the left (white arrow), again only mildly hypermetabolic.

**Figure 10 cancers-14-02835-f010:**
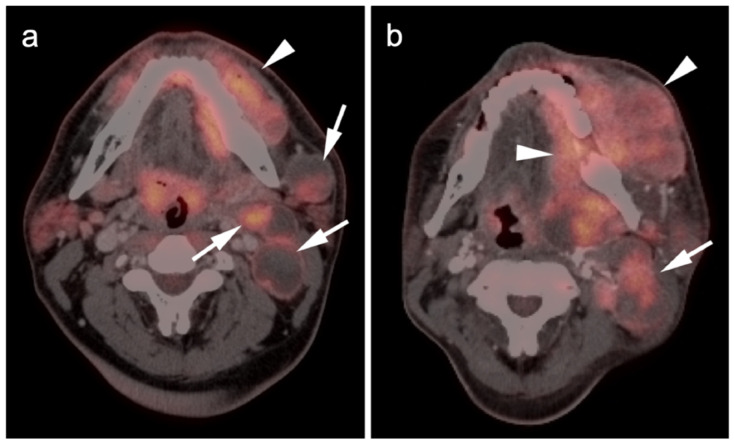
Mucosal melanoma of the oral cavity. Axial FDG PET-CT images of the face. (**a**) At initial diagnosis there is a large, moderately hypermetabolic mass centered in the left submandibular space, medial and lateral to the left mandible (white arrow). There are also multiple involved partially cystic lymph nodes (white arrow). (**b**) At follow-up on treatment 3 months later, there is progressive enlargement of the mass in the left oral cavity (white arrowhead) and persistent nodal disease (white arrow).

**Figure 11 cancers-14-02835-f011:**
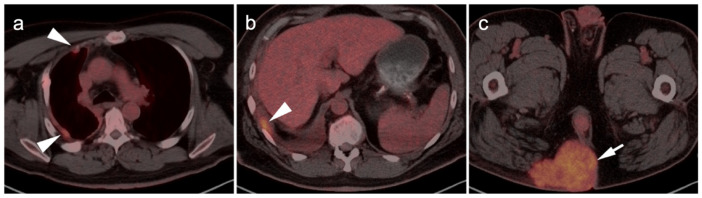
Metastatic Merkel cell carcinoma. (**a**,**b**) Axial FDG PET-CT images of the upper (**a**) and lower (**b**) chest show multiple metastatic hypermetabolic tumor deposits to the right pleura (white arrowheads). (**c**) Axial FDG PET-CT of the low pelvis shows a large primary tumor of the skin and subcutaneous tissue of the right buttock (white arrow).

**Figure 12 cancers-14-02835-f012:**
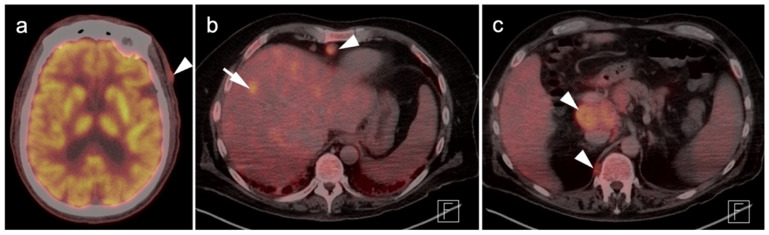
Metastatic Merkel cell carcinoma. (**a**) Axial FDG PET-CT image of the head. The primary tumor in the skin of the left temple (white arrowhead) is small but metabolically active. (**b**) Axial FDG PET-CT of the upper abdomen shows multiple tiny hypermetabolic liver metastases (white arrow. (**c**) Axial FDG PET-CT of the mid-abdomen shows a large portocaval metastatic hypermetabolic nodal conglomerate and a small right retrocrural node (white arrowheads).

**Figure 13 cancers-14-02835-f013:**
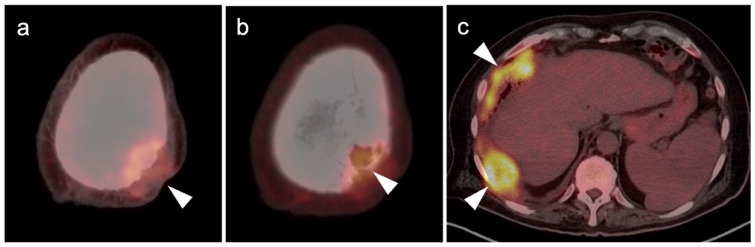
Cutaneous squamous cell carcinoma, metastatic. (**a**) Axial FDG PET-CT of the skull vertex shows a hypermetabolic soft tissue squamous cell carcinoma (white arrowhead) projecting over the skull. (**b**) Axial FDG PET-CT of the skull vertex with bone windows demonstrates destruction of the calvarium (white arrowhead) under the mass. MRI would be needed to evaluate for intracranial extension. (**c**) Axial FDG PET-CT of the upper abdomen shows extensive hypermetabolic right pleural metastases (white arrowheads).

**Figure 14 cancers-14-02835-f014:**
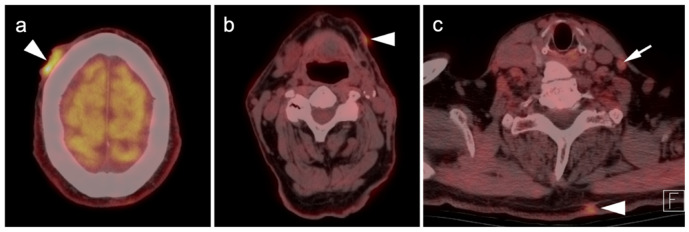
Multiple skin squamous cell carcinomas (SCCs) in a heart transplant patient. (**a**) Axial FDG PET-CT image of the upper head shows a large, ulcerated SCC in the right frontal scalp (white arrowhead). (**b**) Axial FDG PET-CT image of the upper neck shows that an additional SSC is present in the skin under the left jawline (white arrowhead). (**c**) Axial FDG PET-CT image of the lower neck shows a third lesion over the left medial posterior shoulder (white arrow). In addition, a prominent hypermetabolic left level 3/4B lymph node is noted (white arrow).

**Figure 15 cancers-14-02835-f015:**
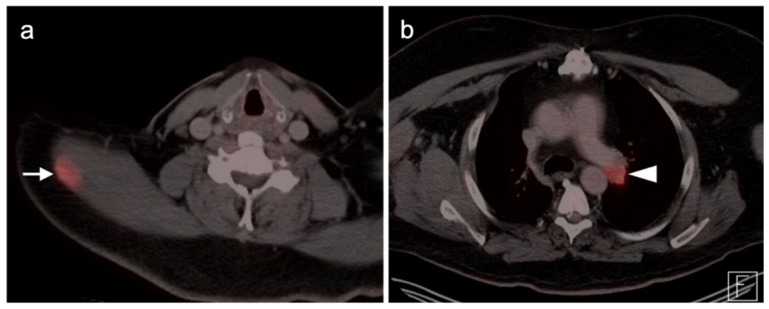
Undifferentiated pleomorphic sarcoma. (**a**) Axial FDG PET-CT image of the low neck shows an intramuscular hypermetabolic mass in the right trapezius muscle (white arrow). (**b**) Axial FDG PET-CT image of the chest shows a hypermetabolic (metastatic) left hilar node (white arrowhead).

**Figure 16 cancers-14-02835-f016:**
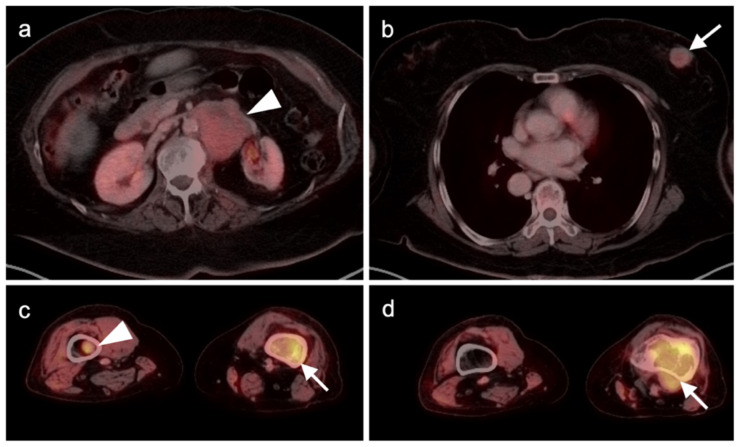
Leiomyosarcoma of the retroperitoneum with late metastatic disease. (**a**) Axial FDG PET-CT image of the abdomen shows a mildly hypermetabolic retroperitoneal mass (white arrowhead) which was known to be present for 4 years before metastasizing. (**b**) Axial FDG PET-CT of the chest shows a hypermetabolic left breast mass, biopsy-proven metastatic leiomyosarcoma (white arrow). (**c**,**d**) Axial FDG PET-CT images of the thighs show metastatic leiomyosarcoma in both femurs (white arrows and arrowhead), with an associated soft tissue mass associated with the left femoral lesion ((**d**), white arrow).

**Figure 17 cancers-14-02835-f017:**
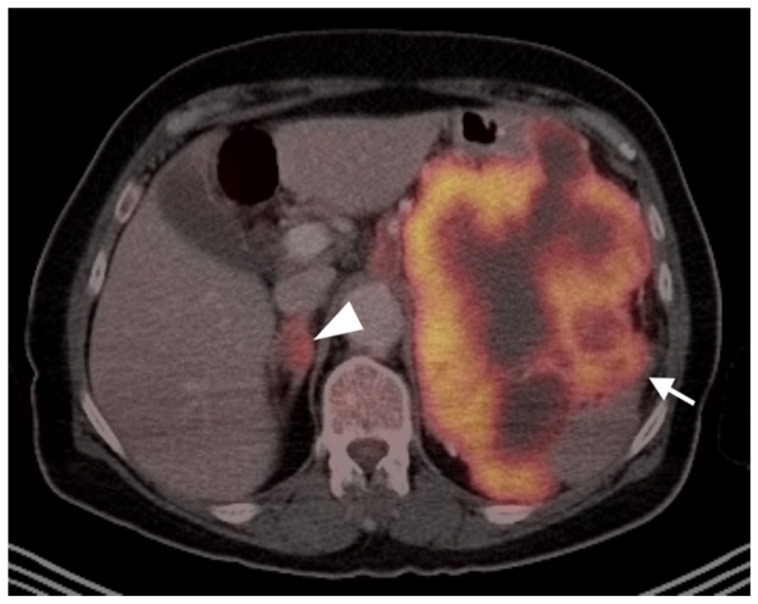
Angiosarcoma of the spleen. Axial FDG PET-CT of the upper abdomen shows an intensely hypermetabolic splenic tumor (white arrow) with scattered areas of necrosis. A hypermetabolic right adrenal metastasis is also present (white arrowhead).

**Figure 18 cancers-14-02835-f018:**
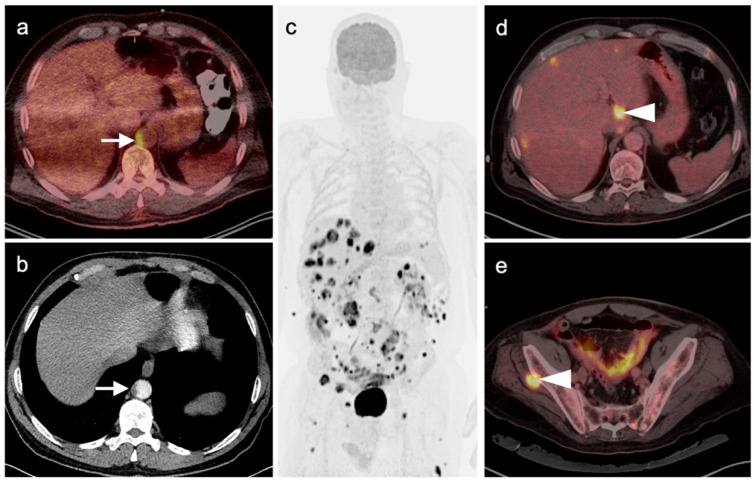
Angiosarcoma of the aorta with downstream hematogenous metastases. (**a**) Axial FDG PET-CT demonstrates a crescentic hypermetabolic lesion in the lower thoracic aorta (white arrow). (**b**) Concurrent axial contrast-enhanced CT shows a crescentic filling defect in the lower thoracic aorta (white arrow). (**c**) Three months later, following radiation of the aortic tumor and an initial thoracic vertebral metastases (not shown) an FDG PET-CT MIP shows the development of numerous metastatic deposits, all “downstream” from the primary aortic lesion. (**d**) Axial FDG PET-CT image of the upper abdomen shows liver metastases (white arrow). (**e**) Axial FDG PET-CT image of the pelvis shows a hypermetabolic lesion in the right gluteus minimus invading the iliac bone (white arrowhead).

**Figure 19 cancers-14-02835-f019:**
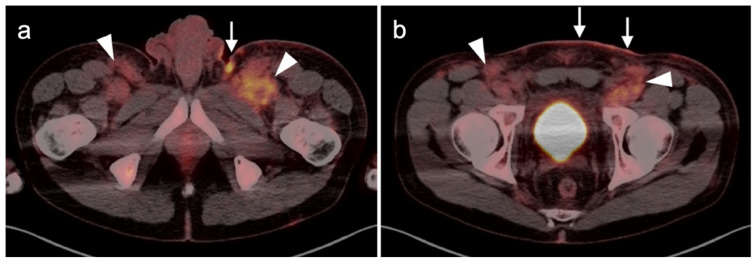
HIV-associated Kaposi sarcoma. (**a**,**b**) Axial FDG PET-CT images of the pelvis demonstrate hypermetabolic cutaneous plaques and nodules (white arrows) and ill-defined hypermetabolic lymph nodes (white arrowheads).

**Figure 20 cancers-14-02835-f020:**
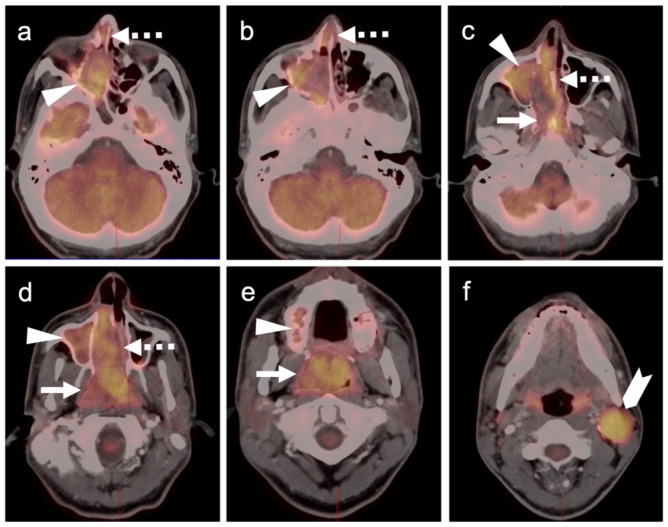
Nasopharyngeal rhabdomyosarcoma. Multiple axial FDG PET-CT images of the head. Extensive and nasopharyngeal rhabdomyosarcoma. (**a**–**e**) Tumor fills the right ethmoid and right maxillary sinuses (white arrowheads); (**c**,**d**) Tumor fills the nasal cavity (dashed white arrows); (**c**,**d**) Tumor also fills the nasopharynx (white arrow); (**f**) There is also involvement of a left level 2 enlarged cervical node (white chevron).

**Figure 21 cancers-14-02835-f021:**
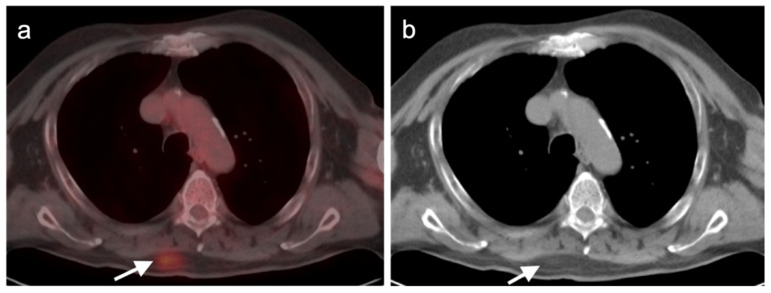
Hibernoma (benign brown fat lipoma). (**a**) Axial FDG PET-CT image of the chest shows an incidental hypermetabolic (SUVmax 5.3) fat-attenuating region in the right posterior paraspinous region as a small, mobile, slightly firm nodule (white arrow). (**b**) A concurrent axial CT image of the chest shows the fat to be minimally high in attenuation at this site (white arrow). Biopsy showed adipocytes and brown fat cells with hypervascularity. There was no evidence of malignancy. The appearance would be indistinguishable from low-grade liposarcoma on FDG PET-CT imaging.

**Figure 22 cancers-14-02835-f022:**
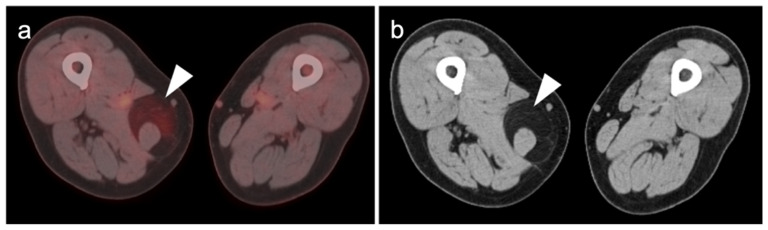
Low-grade liposarcoma. (**a**,**b**) Axial FDG PET-CT (**a**) and a concurrent axial CT image (**b**) of the thigh show a fatty tumor surrounding the sartorius muscle of the right thigh with a region of increased metabolic activity (white arrowheads) as well as a more posterior region where metabolic activity is low. The patient had pain in the mass, which at biopsy was found to be a low-grade liposarcoma.

**Figure 23 cancers-14-02835-f023:**
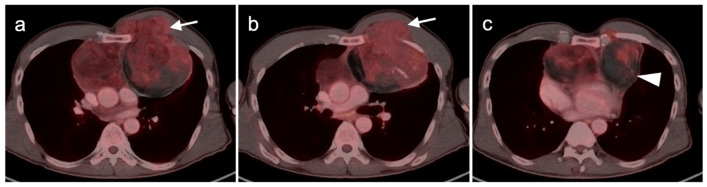
Dedifferentiated liposarcoma of the mediastinum arising from a low-grade liposarcoma. A large anterior mediastinal/chest wall mass had been present for many years but had recently increased in size and become symptomatic. (**a**–**c**) Axial FDG PET-CT images of the chest show a heterogenous mass in the anterior mediastinum with invasion of the left anterior chest wall (white arrows). (**c**) Hypometabolic regions of low attenuation (white arrowhead) were also present. Metabolically active regions were targeted at biopsy, revealing dedifferentiated liposarcoma arising from a low-grade liposarcoma.

**Figure 24 cancers-14-02835-f024:**
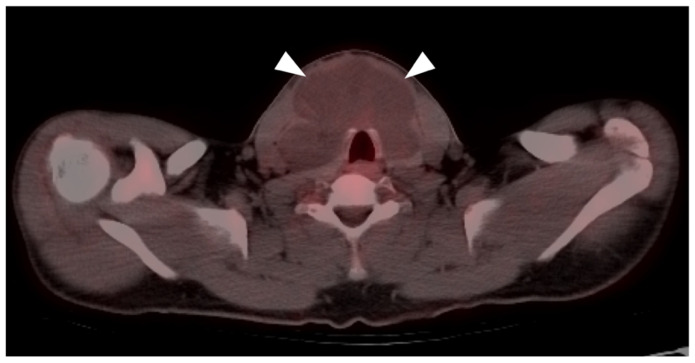
Myxoid liposarcoma of the central compartment of the neck. On an axial FDG PET-CT image of the neck, a cystic tumor in the central compartment of the anterior neck was observed to be mildly metabolically active and higher in attenuation than most lipomas (white arrows). Myxoid liposarcomas can be multi-loculated on MRI.

**Figure 25 cancers-14-02835-f025:**
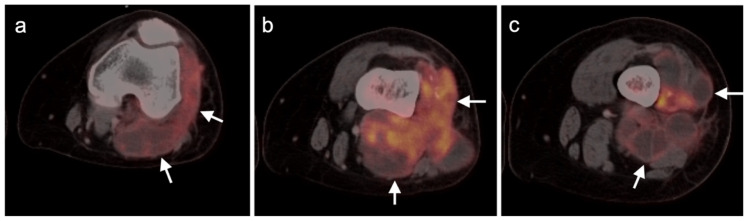
Synovial cell sarcoma of the knee. Axial FDG PET-CT images of the left lower thigh. (**a**) There is a lobulated hypermetabolic mass arising from the synovial capsule of the left knee (white arrow). (**b**,**c**) A lobulated, hypermetabolic mass containing multi-loculated cystic areas dissects into the surrounding soft tissues (white arrows).

**Figure 26 cancers-14-02835-f026:**
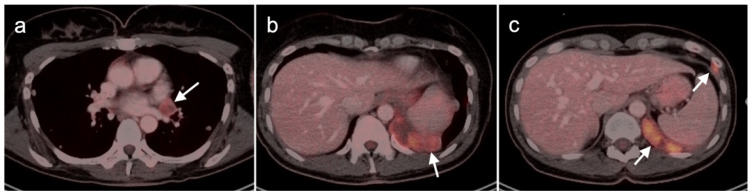
Primary synovial sarcoma of the pleura. (**a**–**c**) There are multiple hypermetabolic masses in the left pleural space shown on axial FDG PET-CT images of the chest (white arrows). No other sites of tumor were identified.

**Figure 27 cancers-14-02835-f027:**
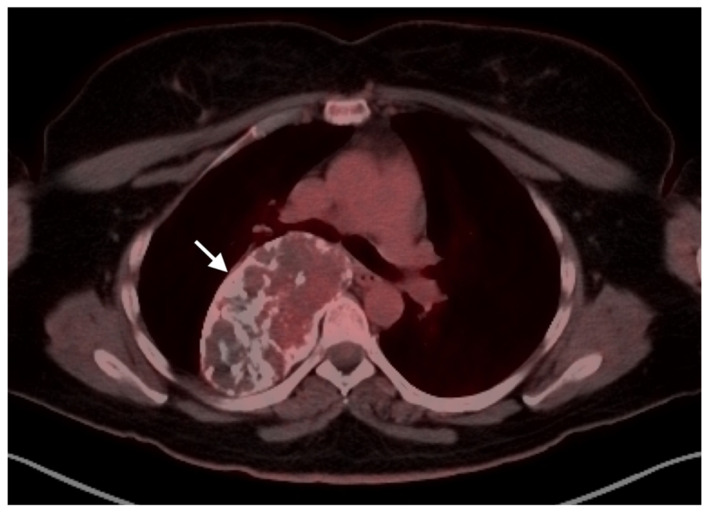
Ganglioneuroma. Axial FDG PET-CT image of the chest shows a large, partially calcified tumor of the right posterior chest (white arrow). These tumors arise from intercostal nerves that connect to sympathetic thoracic ganglia and are typically benign.

**Figure 28 cancers-14-02835-f028:**
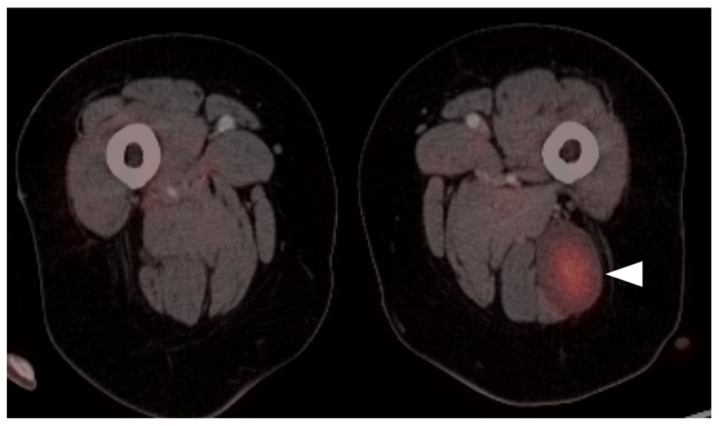
Low-grade malignant nerve sheath tumor. Features that suggested that the lesion (white arrowhead) could be malignant on this axial FDG PET-CT image of the thighs was an SUVmax of 6.8 (in the indeterminate range), an enlarging mass and sensory symptoms (described as a “tingling, burning sensation”).

**Figure 29 cancers-14-02835-f029:**
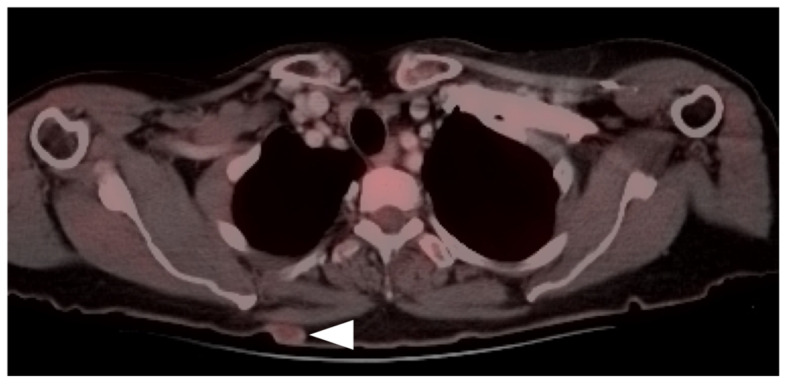
Cutaneous neurofibroma in a patient with neurofibromatosis type-1 (NIF-1). The nodule showed very mild metabolic activity (white arrowhead) similar to that of background muscle.

**Figure 30 cancers-14-02835-f030:**
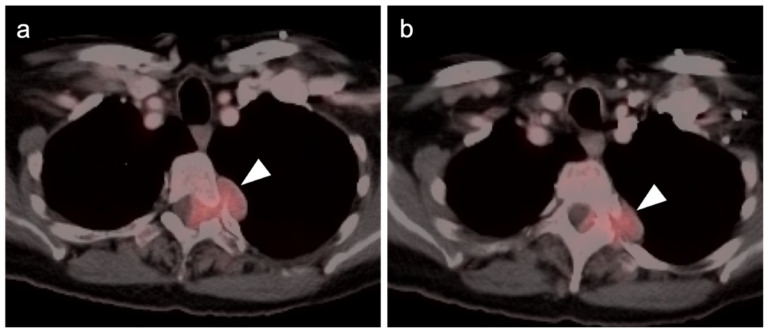
Dumbbell neurofibroma of the thoracic spine. (**a**,**b**) Axial FDG PET-CT images of the chest show a dumbbell-shaped lesion with an epidural/intraspinous component extending through the lateral recess to an extraspinous component (white arrowheads). As in this case, patients often experience pain and symptoms of spinal cord compression. The lesion was surgically removed and found to be a benign neurofibroma.

**Figure 31 cancers-14-02835-f031:**
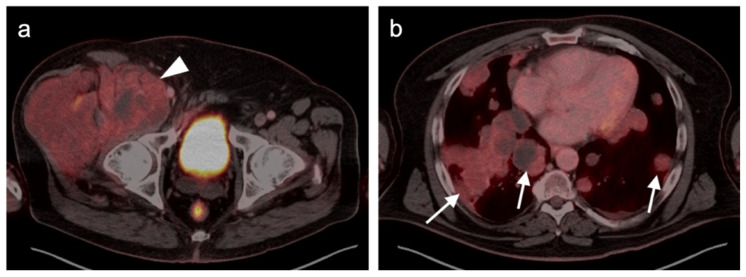
Myxoid chondrosarcoma. (**a**) An axial FDG PET-CT image of the pelvis demonstrates a large, moderately hypermetabolic soft tissue mass arising between muscle groups of the right anterior pelvis (white arrowhead). (**b**) An axial FDG PET-CT image of the chest shows multiple hypermetabolic lung metastases (white arrows), some with central necrosis.

**Figure 32 cancers-14-02835-f032:**
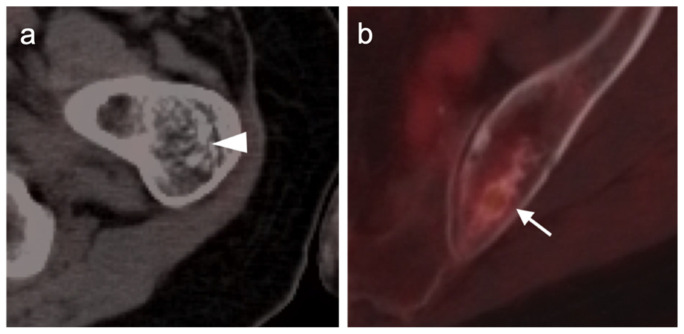
Variable appearance of benign enchondromas on FDG PET-CT. (**a**) A magnified view of an axial FDG PET-CT image of the left proximal femur demonstrates a non-metabolically active, centrally calcified left intertrochanteric lesion (white arrowhead). (**b**) A magnified view of an axial FDG PET-CT image of the pelvis shows mild metabolic activity in a centrally calcified lesion of the medial left iliac bone (white arrow). Both were unchanged over many years and asymptomatic. The development of pain referable to what appears to be a benign enchondroma should prompt evaluation for conversion to possible low-grade chondrosarcoma. Some benign enchondromas can show mild metabolic activity on FDG PET.

**Figure 33 cancers-14-02835-f033:**
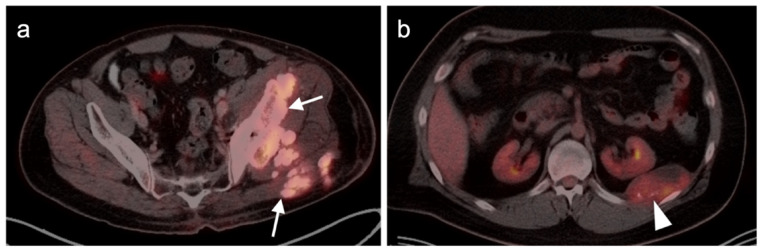
Osteosarcoma of the left iliac bone. (**a**) On an axial FDG PET-CT image, a bone-forming hypermetabolic tumor of the left iliac bone is associated with hypermetabolic, ossified, adjacent intramuscular metastases (white arrows). (**b**) An axial FDG PET-CT of the abdomen shows a pleural hypermetabolic metastatic lesion in the deep left posterior costophrenic sulcus with scattered internal calcifications (white arrowhead).

**Figure 34 cancers-14-02835-f034:**
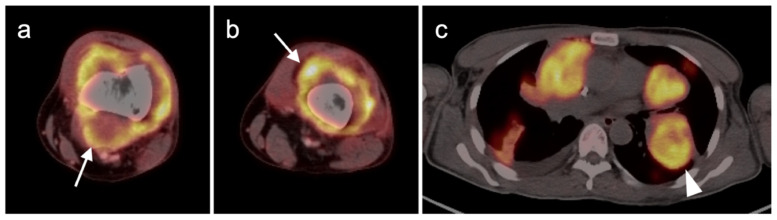
Osteosarcoma of the left distal femur with lung metastases. (**a**,**b**) Axial FDG PET-CT of the left distal femur shows a hypermetabolic mass (white arrows) primarily within the soft tissues surrounding the sclerotic bone. The patient had recently completed neoadjuvant chemoradiation therapy (CRT) for an osteosarcoma of the left distal femur, with poor treatment response. (**c**) Multiple cannon ball, intensely hypermetabolic lung metastases (white arrowheads) are present on an axial FDG PET-CT of the chest. Pre-CRT, the lung metastases were very small.

**Figure 35 cancers-14-02835-f035:**
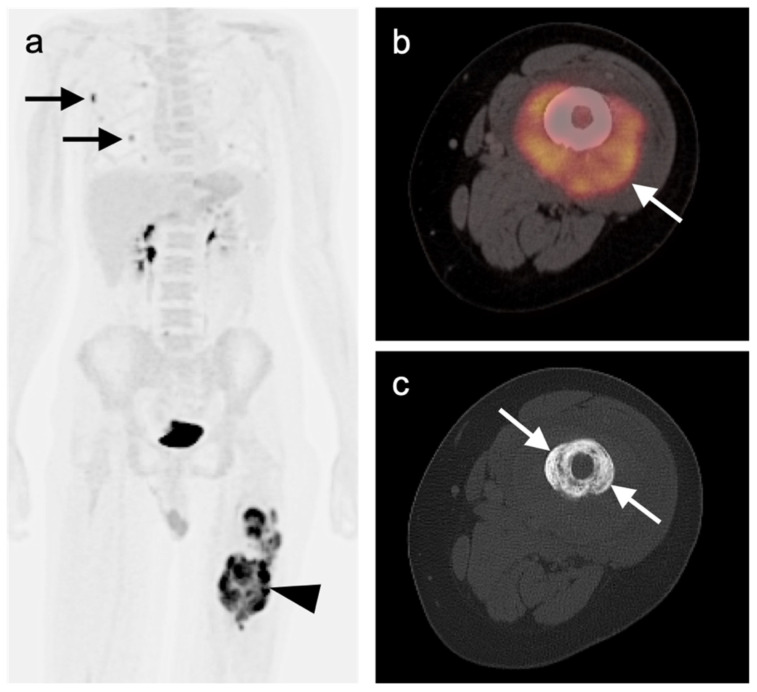
Ewing sarcoma of the left femur. (**a**) FDG PET MIP image demonstrates an intensely hypermetabolic left femoral mass (black arrowhead) and multiple small hypermetabolic lung nodules (black arrows). (**b**) Axial FDG PET-CT image of the left femur demonstrates a circumferential hypermetabolic soft tissue mass (white arrow) in the left femoral diaphysis, with metabolic activity extending through the bone to the medullary space. (**c**) Axial CT image of the left femur demonstrates lamellated concentric (onion-skin) cortical thickening, typical of Ewing sarcoma (white arrows).

**Figure 36 cancers-14-02835-f036:**
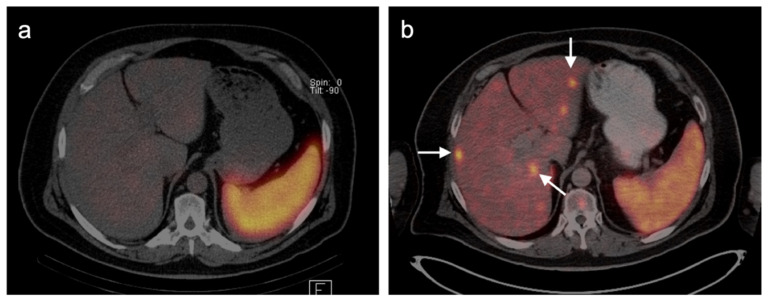
Metastatic carcinoid in the liver: comparison of [^111^In]In-pentetreotide (OctreoScan^®^) SPECT-CT and [^64^Cu]Cu-DOTA-TATE (Detectnet^®^) PET-CT. (**a**) An axial OctreoScan^®^ SPECT-CT image of the upper abdomen shows no obvious liver metastases. (**b**) An axial Detectnet^®^ PET-CT scan performed a short time later shows multiple focal PET-positive lesions within the liver white arrows) and a single vertebral lesion that were not seen on Octreoscan^®^ SPECT-CT. This illustrates the difference in sensitivity between OctreoScan^®^ SPECT-CT and SSTR PET-CT.

**Figure 37 cancers-14-02835-f037:**
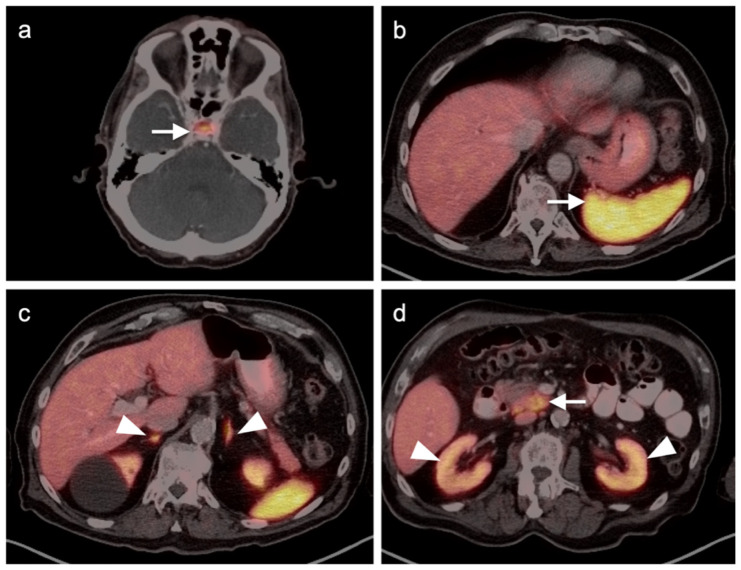
Normal biodistribution of [^68^Ga]Ga-DOTA-TATE (NETSPOT^®^) PET-CT. (**a**) Normal uptake in the pituitary gland on an axial PET-CT image of the head (white arrow). (**b**) Normal intense uptake in the spleen on an axial PET-CT image of the upper abdomen (white arrow). (**c**) Normal uptake in the adrenal glands on an axial PET-CT image of the abdomen (white arrowheads) (note incidental right renal cyst). (**d**) Normal uptake on an axial PET-CT image of the abdomen in the uncinate process of the pancreas (white arrow) and in the kidneys (white arrowheads). A similar biodistribution is also seen with [^64^Cu]Cu-DOTA-TATE (Detectnet^®^).

**Figure 38 cancers-14-02835-f038:**
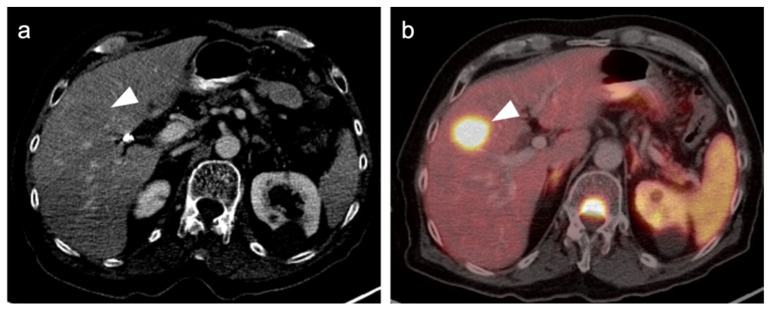
Comparison between contrast-enhanced CT and [^68^Ga]Ga-DOTA-TATE (NETSPOT^®^) PET-CT in the detection of a hepatic carcinoid metastasis. (**a**) Axial contrast-enhanced CT image of the liver shows poor definition of a metastatic well-differentiated neuroendocrine tumor (white arrowhead). (**b**) Axial [^68^Ga]Ga-DOTA-TATE PET-CT shows an intensely PET-positive metastasis (white arrowhead), suggesting well-differentiated histology. There is also a metastasis to the adjacent vertebral body. There are several left renal cysts.

**Figure 39 cancers-14-02835-f039:**
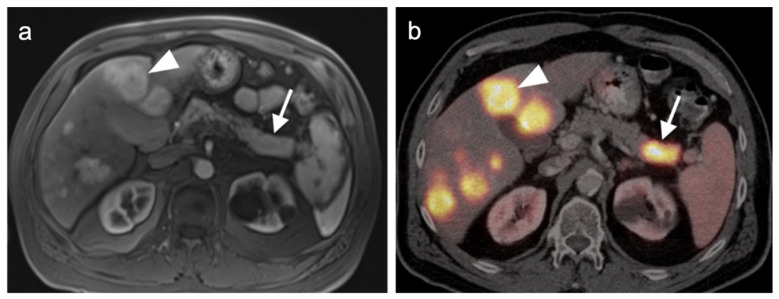
Comparison between MRI and SSTR PET-CT in the detection of primary pancreatic neuroendocrine tumor and hepatic metastases. (**a**) Axial Gd-enhanced T1 FS MRI image of the upper abdomen demonstrates increased signal in multiple hepatic metastases (white arrowhead) and in the primary lesion in the tail of the pancreas (white arrow). (**b**) Axial [^68^Ga]Ga-DOTA-TATE PET-CT through the same regions shows intensely PET-positive lesions in the liver (white arrowhead) and the tail of the pancreas (white arrow), with similar detection to MRI.

**Figure 40 cancers-14-02835-f040:**
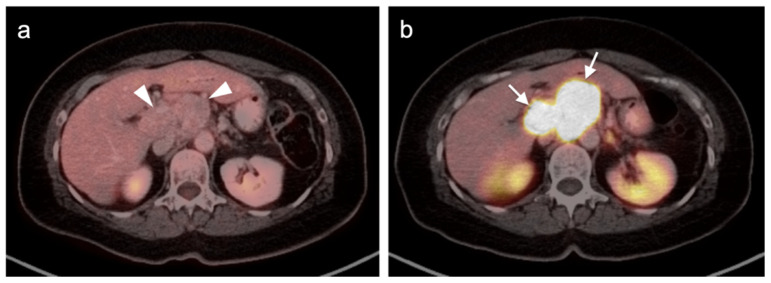
Well-differentiated NET comparison between FDG PET-CT (**a**) and [^68^Ga-DOTA-TATE PET-CT (**b**). (**a**) Axial FDG PET-CT of the upper abdomen in a patient with a well-differentiated carcinoid tumor metastatic to the caudate lobe of the liver and bulky nodes at the hepatic hilum showing low metabolic activity on FDG PET-CT (white arrowheads). (**b**) An axial [^68^Ga]-DOTA-TATE PET-CT image for this same patient shows intense activity in the metastatic mass (white arrows). Based on striking uptake of DOTATATE in the metastatic lesions, this patient might be a good candidate for Lutathera^®^.

**Figure 41 cancers-14-02835-f041:**
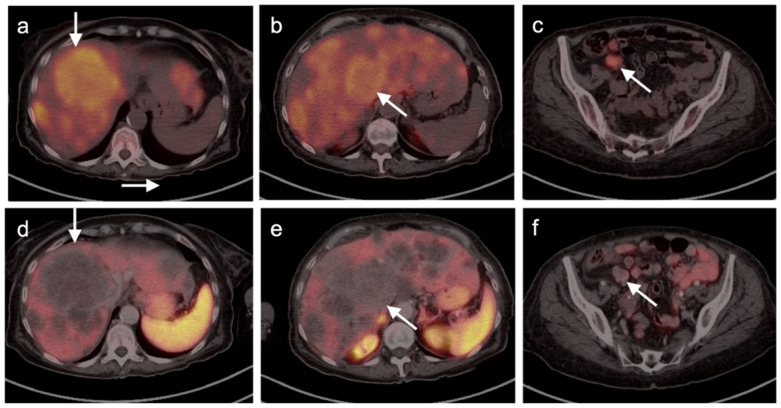
Comparison between FDG PET-CT (**a**–**c**) and [^68^Ga]Ga-DOTA-TATE PET-CT (**d**–**f**) in a patient with poorly differentiated NET. (**a**–**c**), Axial FDG PET-CT images of the abdomen and upper pelvis show multiple hypermetabolic liver metastases (white arrows) and a hypermetabolic mesenteric mass (white arrow). (**d**–**f**) Axial [^68^Ga]Ga-DOTA-TATE PET-CT images of the abdomen and upper pelvis show multiple liver metastases and the mesenteric mass with no demonstrable uptake (white arrows). This patient would not be a good candidate for Lutathera^®^.

**Figure 42 cancers-14-02835-f042:**
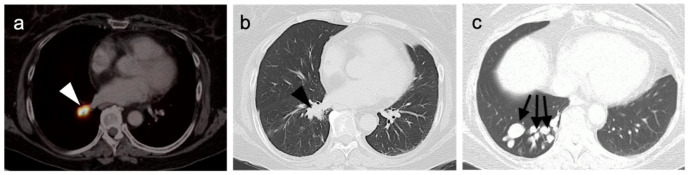
Typical pulmonary carcinoid and associated bronchoceles. (**a**) Axial [^64^Cu]Cu-DOTA-TATE (Detectnet^®^) PET-CT shows the markedly PET-positive bronchial lesion in the right infrahilar region (white arrowhead), consistent with a well-differentiated (typical) bronchial carcinoid tumor. (**b**) Axial CT shows the location of the lesion that was positive on PET-CT (black arrowhead). (**c**) Axial CT shows bronchoceles (black arrows) related to dilated fluid-filled airways, a frequent finding with pulmonary carcinoids (these bronchoceles showed no uptake on SSTR PET-CT).

**Figure 43 cancers-14-02835-f043:**
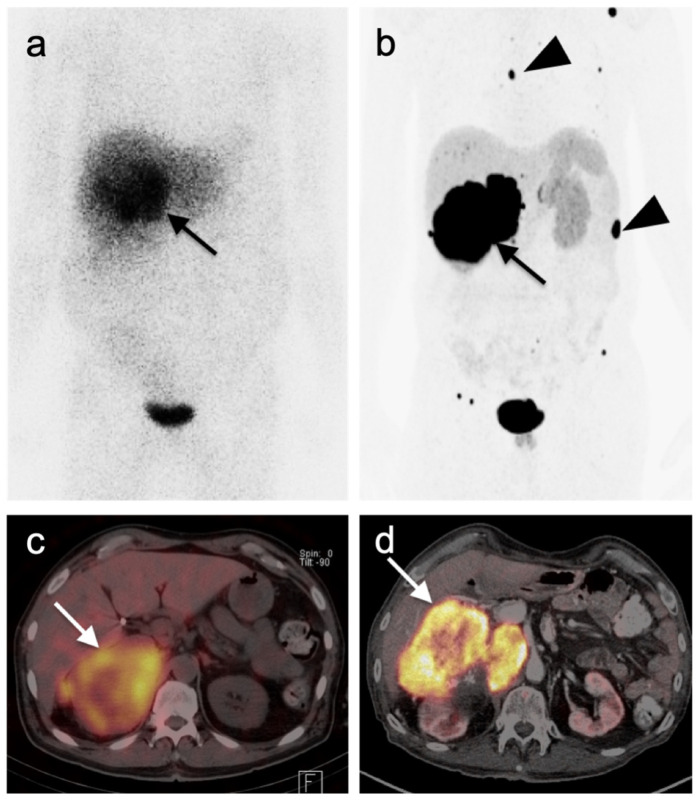
Pheochromocytoma. ((**a**,**c**), left panels) [^123^I]I-MIBG. Anterior planar imaging of the chest and abdomen (**a**) and an axial SPECT/CT of the upper abdomen with an MIBG scan shows prominent uptake in a large right pheochromocytoma (white arrows). ((**b**,**d**), right panels) [^64^Cu]Cu-DOTA-TATE (Detectnet^®^ PET-CT performed several years later in the same patient. (**b**) Upper right is a MIP image of the PET-CT scan, and the right lower image is the corresponding axial PET-CT image showing intense uptake in the mass, which has increased in size (arrows). The patient also has scattered additional foci of activity on the MIP PET image (representative lesions with black arrowheads) that were bone metastases.

**Figure 44 cancers-14-02835-f044:**
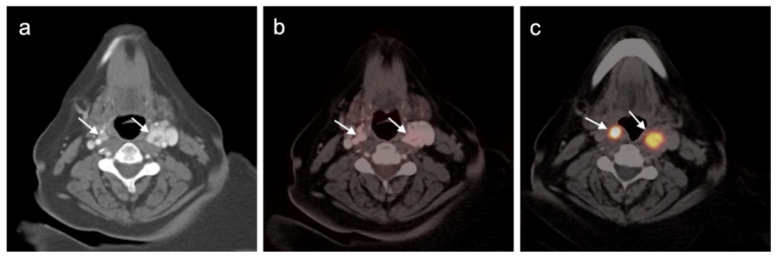
Bilateral carotid body paragangliomas with comparison contrast-enhanced CT and FDG and [^68^Ga]Ga-DOTA-TATE (NETSPOT^®^) PET-CT scans. (**a**) Axial intravenous contrast-enhanced CT image of the upper neck shows strongly enhancing bilateral nodules in the carotid space at the carotid bifurcation (white arrows). (**b**) Axial image of the upper neck with a concurrently acquired FDG PET-CT shows that the nodules exhibit relatively low metabolic activity (white arrows). (**c**) Axial [^68^Ga]Ga-DOTA-TATE (NETSPOT^®^) image of the upper neck shows that the nodules demonstrate intense activity (white arrows). Paragangliomas are reliably strongly positive on SSTR PET-CT but are variable in terms of metabolic activity on FDG PET-CT.

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
