# Peer review of "PET-CT in Clinical Adult Oncology—VI. Primary Cutaneous Cancer, Sarcomas and Neuroendocrine Tumors"

_cancers, 2022, doi:10.3390/cancers14122835_

Round 1

Reviewer 1 Report

In this work the Authors aim at providing an overview of the value, applications, imaging and interpretive strategies of PET-CT in primary cutaneous cancer, sarcomas and neuroendocrine tumors. The review is comprehensive and concise and could help the reader to get a quick but sufficient insight on the topic of interest, that it may also be not so newly discovered, but it is surely up-to-date.

Minor revisions

  • Please uniform in all the manuscript the words “NETs” and “NETS”
  • Line 832: please check the reference “[642]”
  • Lines 962-963: I suggest including the following works:

https://doi.org/10.3390/diagnostics11020192 on the role of dual tracer 68Ga-DOTATOC and 18F-FDG PET in the improvement of preoperative evaluation of aggressiveness in resectable pancreatic NETs

https://doi.org/10.1007/s40336-019-00328-1 on the role of combined 68Ga DOTA-peptides and 18F-FDG PET in the diagnosis and management of patients with NEN.

Author Response

Reviewer 1 comments (responses in red)

In this work the Authors aim at providing an overview of the value, applications, imaging and interpretive strategies of PET-CT in primary cutaneous cancer, sarcomas and neuroendocrine tumors. The review is comprehensive and concise and could help the reader to get a quick but sufficient insight on the topic of interest, that it may also be not so newly discovered, but it is surely up-to-date. Thank you.

Minor revisions

1. Please uniform in all the manuscript the words “NETs” and “NETS”.

We have corrected all of these to “NETs”.

2. Line 832: please check the reference “[642]”.

The reference number has been corrected.

3. Lines 962-963: I suggest including the following works:

https://c on the role of dual tracer 68Ga-DOTATOC and 18F-FDG PET in the improvement of preoperative evaluation of aggressiveness in resectable pancreatic NETs

https://doi.org/10.1007/s40336-019-00328-1 on the role of combined 68Ga DOTA-peptides and 18F-FDG PET in the diagnosis and management of patients with NET.

We have revised this section of the manuscript and have also included these helpful references as well as some others.

Reviewer 2 Report

-for Merkel cell tumors, the use of somatostatin radiopharmaceuticels should be mentioned

-the significance of sentinental node scintigraphy should be covered for evaluating equivocal PET/CT scans, mainly in Malignant Melanoma (Ahmadzadehfar and Hinz)

- one of the early PET/CT papers on Malignant Melanoma bei Reinhardt M (J Clin Oncol 2008) should be cited.

Author Response

Reviewer 2 comments (responses in red)

-for Merkel cell tumors, the use of somatostatin radiopharmaceuticels should be mentioned.

We have included this in the revised manuscript and included a reference.

-the significance of sentinental node scintigraphy should be covered for evaluating equivocal PET/CT scans, mainly in Malignant Melanoma (Ahmadzadehfar and Hinz).

We have included a discussion of SNL scintigraphy as it pertains to FDG PET-CT this in the revised manuscript. We have included the reference provided.

- one of the early PET/CT papers on Malignant Melanoma bei Reinhardt M (J Clin Oncol 2008) should be cited.

We have included a discussion of this and cited the reference in the revised manuscript.

Reviewer 3 Report

Thank you very much for the opportunity to review the article entitled: “PET-CT in clinical adult oncology – VI. Primary cutaneous cancer, sarcomas and neuroendocrine tumors.” Event thus this article is interesting, some major changes are required before publication:

  • In a lots of place in the manuscript authors does not provide appropriate references. Whenever they indicate the numbers or % this should be supported with the appropriate references. For example in the introduction section there is no references as well as in the 2.1 section.
  • A majority of the abbreviation does not have an explanation, for example (but not limited to): MR, PPV, NPV, NCCN, ECORT, WHO, DWI, GU, CRT; some of abbreviations are used twice for example: irAEs, some are missing like non-attenuation corrected.
  • The figures are not homogenous: once “a” or “b” are on the image, once are below. Moreover, figures with two rows should look like in the Figure 2.
  • I recommend to provide a major stylistic changes in the whole manuscript.
  • What ‘review’ reference in the main manuscript means?
  • In the section 2.1, line 89-90: please indicate what “quite rare” means. Preferable with the number when authors compare AC of the skin with the other tumors
  • In section 2.3 Melanoma, section “1 Cutaneous melanoma” it should be section 2.3.1 not 1 (it looks like it is the section belonging to the Introduction. The same in the other section. Please provide changes – this is misleading the reader.
  • Figure 8: “note that not all vulvar melanomas are this hypermetabolic” – this should not be in the figure description, it should be in the main text;
  • Page 14, section 2, line 475 – where “Section 2.5.3” which author refer to is in the main text?
  • Page 15 figure 16 – could authors provide an appropriate images to confirm that “size of the tumor was unchanged for the 4 years’? If not please delete this information, without a confirmation this information is not scientific
  • Figure 20 and 21 – please add A-F and add arrow or arrowheads like in other figures.
  • Page 14, section 2, line 480-482 – please indicate values for SUV and MTV for which authors refer to.
  • Page 15, section 3, lines 503-503 - please indicate values for SUV and MTV for which authors refer to.
  • Page 17, section 5, lines 559-560: please indicate values for MTV for which authors refer to.
  • Page 21, lines 657-663 – this information is out of scope of this manuscript because it is referred to MRI imaging, not PET/CT
  • Page 23, section 1, line 696: “myxoid chondrosarcomas arise in the soft tissue rather than bone, however (Figure 31).” - please provide appropriate stylistic and linguistic changes in this sentence as well as in the whole manuscript.
  • Why authors does not add information about NaF or PSMA in the imaging of sarcomas? The title of this manuscript is “PET/CT in Clinical Adult Oncology”, so it suggest that they will discuss different radiotracers, not only [18F]FDG. Moreover authors indicate that they will discuss several radiotracers. There is an proven evidence in the literature that NaF and PSMA has a diagnostic accuracy in sarcoma imaging.
  • Page 25, section 2, line 759-761: does author are able to provide appropriate numbers/values for confirmation of this sentence?
  • Page 27, section 4.1, line 832 – where reference “642” is?
  • Page 30, section 4.3, line 935 – where figure 205 is?
  • Please provide changes of the nomenclature in whole manuscript according to "Consensus nomenclature rules for radiopharmaceutical chemistry — setting the record straight.
  • SUVmax is usually written to one decimal place.
  • Please provide changes in the reference section according to the journal guideline.

Minor changes:

Please check carefully the whole manuscript for typos, double spacer or missing dots for example (but not limited to): page 34 line 1023: Sharma et al – the dot is missing

Author Response

Reviewer 3 comments (responses in red)

Thank you very much for the opportunity to review the article entitled: “PET-CT in clinical adult oncology – VI. Primary cutaneous cancer, sarcomas and neuroendocrine tumors.” Event thus this article is interesting, some major changes are required before publication:

  • In a lots of place in the manuscript authors does not provide appropriate references. Whenever they indicate the numbers or % this should be supported with the appropriate references. For example in the introduction section there is no references as well as in the 2.1 section.
  • We have corrected this and have provided appropriate references
  • A majority of the abbreviation does not have an explanation, for example (but not limited to): MR, PPV, NPV, NCCN, ECORT, WHO, DWI, GU, CRT; some of abbreviations are used twice for example: irAEs, some are missing like non-attenuation corrected.
  • We have correctly defined all abbreviations.
  • The figures are not homogenous: once “a” or “b” are on the image, once are below. Moreover, figures with two rows should look like in the Figure 2.
  • We have completely reformatted the figures to include the labeling embedded in the figures, as suggested, to look like Figure 2.
  • I recommend to provide a major stylistic changes in the whole manuscript. We have attempted to implement this by extensively rewriting the manuscript.
  • What ‘review’ reference in the main manuscript means?
  • It denoted that this is was review article., but we have removed this throughout.
  • In the section 2.1, line 89-90: please indicate what “quite rare” means. Preferable with the number when authors compare AC of the skin with the other tumors.
  • We have removed this and have focused on the more common skin cancers.
  • In section 2.3 Melanoma, section “1 Cutaneous melanoma” it should be section 2.3.1 not 1 (it looks like it is the section belonging to the Introduction. The same in the other section. Please provide changes – this is misleading the reader.
  • We have corrected this.
  • Figure 8: “note that not all vulvar melanomas are this hypermetabolic” – this should not be in the figure description, it should be in the main text; We have corrected this.
  • Page 14, section 2, line 475 – where “Section 2.5.3” which author refer to is in the main text?
  • We removed this. This was addressed in another review article in this series.
  • Page 15 figure 16 – could authors provide an appropriate images to confirm that “size of the tumor was unchanged for the 4 years’? If not please delete this information, without a confirmation this information is not scientific.
  • We have clarified this.
  • Figure 20 and 21 – please add A-F and add arrow or arrowheads like in other figures.
  • This has been corrected.
  • Page 14, section 2, line 480-482 – please indicate values for SUV and MTV for which authors refer to.
  • The images used were from a deidentified teaching file and therefore cannot be accessed for additional measurements. Visually, the lesions are significantly hypermetabolic compared to background tissues, which has been clarified.
  • Page 15, section 3, lines 503-503 - please indicate values for SUV and MTV for which authors refer to.
  • The images used were from a deidentified teaching file and therefore cannot be accessed for additional measurements. Visually, the lesions are significantly hypermetabolic compared to background tissues, which has been clarified.
  • Page 17, section 5, lines 559-560: please indicate values for MTV for which authors refer to.
  • The images used were from a deidentified teaching file and therefore cannot be accessed for additional measurements. Visually, the lesions are significantly hypermetabolic compared to background tissues, which has been clarified.
  • Page 21, lines 657-663 – this information is out of scope of this manuscript because it is referred to MRI imaging, not PET/CT.
  • This has been clarified as relevant for correlative imaging.
  • Page 23, section 1, line 696: “myxoid chondrosarcomas arise in the soft tissue rather than bone, however (Figure 31).” - please provide appropriate stylistic and linguistic changes in this sentence as well as in the whole manuscript.
  • This sentence has been improved and we have attempted to review the entire manuscript with respect to stylistic and linguistic consistency.
  • Why authors does not add information about NaF or PSMA in the imaging of sarcomas? The title of this manuscript is “PET/CT in Clinical Adult Oncology”, so it suggest that they will discuss different radiotracers, not only [18F]FDG. Moreover authors indicate that they will discuss several radiotracers. There is an proven evidence in the literature that NaF and PSMA has a diagnostic accuracy in sarcoma imaging.
  • We have added a discussion of NaF and PSMA in sarcoma imaging.
  • Page 25, section 2, line 759-761: does author are able to provide appropriate numbers/values for confirmation of this sentence?
  • This has been provided.
  • Page 27, section 4.1, line 832 – where reference “642” is?
  • This has been corrected.
  • Page 30, section 4.3, line 935 – where figure 205 is?
  • This has been corrected.
  • Please provide changes of the nomenclature in whole manuscript according to "Consensus nomenclature rules for radiopharmaceutical chemistry — setting the record straight.
  • We have proof-read the manuscript, making appropriate changes to nomenclature.
  • SUVmax is usually written to one decimal place.
  • We have made corrections, accordingly.
  • Please provide changes in the reference section according to the journal guideline.
  • The references were provided in NML format throughout. The editors will convert this to the journal reference style.

Minor changes:

Please check carefully the whole manuscript for typos, double spacer or missing dots for example (but not limited to): page 34 line 1023: Sharma et al – the dot is missing.

We carefully reread the manuscript and make corrections as suggested.

Reviewer 4 Report

This manuscript provides a comprehensive review of PET-CT in three major cancer types with a focus on FDG and SSTR radiotracers. It is well written with a lot of useful information. Some other comments are as below:

1.         Please spell out all abbreviations.

2.         Please list the references in the legends if the figure is from other literature.

3.         Please list the specific radiotracer names for the figures done with SSTR PET since both 68Ga and 64Cu versions of DOTATATE are available. It is not the same case for FDG PET.

4.         There are some minor errors in writing. Please read through and correct when needed. For example:

Abstract, line 39: “this article focuses on the most common of these malignancies…”.

Page 2, line 76: “the focus herein is on PET-CT because if its widespread availability”

Page 4, line 157: “it critical that a baseline FDG”

5.         Line 1026: “131” should be in superscript format.

6.         Page 5, line 191: please change “1. Cutaneous melanoma” to “2.3.1 Cutaneous melanoma”. Please follow the same recommendations for other similar parts.

Author Response

Reviewer 4 comments (responses in red)

This manuscript provides a comprehensive review of PET-CT in three major cancer types with a focus on FDG and SSTR radiotracers. It is well written with a lot of useful information.

Thank you.

Some other comments are as below:

  1. Please spell out all abbreviations.

We have corrected this in the revised manuscript.

  1. Please list the references in the legends if the figure is from other literature.

All figures are original.

  1. Please list the specific radiotracer names for the figures done with SSTR PET since both 68Ga and 64Cu versions of DOTATATE are available. It is not the same case for FDG PET.

We have made this correction.

  1. There are some minor errors in writing. Please read through and correct when needed. For example:

Thank you. We have corrected the following errors in the revised manuscript and have carefully reread the manuscript for writing errors.

  • Abstract, line 39: “this article focuses on the most common of these malignancies…”.
  • Corrected
  • Page 2, line 76: “the focus herein is on PET-CT because if its widespread availability”
  • Corrected
  • Page 4, line 157: “it critical that a baseline FDG”.
  • Corrected
  • Line 1026: “131” should be in superscript format.
  • Corrected
  1. Page 5, line 191: please change “1. Cutaneous melanoma” to “2.3.1 Cutaneous melanoma”. Please follow the same recommendations for other similar parts.

We have reviewed the manuscript and have corrected this throughout.

Round 2

Reviewer 3 Report

Thank you very much for provided majority of changes into manuscript entiteled: “PET-CT in clinical adult oncology – VI. Primary cutaneous cancer, sarcomas and neuroendocrine tumors.” However, still some minor changes are required:

1)      Not all abbreviation Authors explained for example (but not limited to): MR, FDG, paragraph 2.2.

2)      Please indicate homogenous nomenclature for imaging modalities: once is MR once is MRI.

3)      Please provide changes of the nomenclature in whole manuscript according to "Consensus nomenclature rules for radiopharmaceutical chemistry — setting the record straight.  – still the radiopharmaceutical nomenclature is wrong written. It’s not only to FDG but also to PSMA or [18F]NaF – please provide appropriate changes.

4)      SUVmax is usually written to one decimal place. – this was not changed

Author Response

Round 2 review (authors reply in red)

1)    Not all abbreviation Authors explained for example (but not limited to): MR, FDG, paragraph 2.2.

We have carefully read the manuscript and we think we have found and defined all the abbreviations the first time they were used.

2)    Please indicate homogenous nomenclature for imaging modalities: once is MR once is MRI.

We have changed all MR to MRI.

3)    Please provide changes of the nomenclature in whole manuscript according to "Consensus nomenclature rules for radiopharmaceutical chemistry — setting the record straight.  – still the radiopharmaceutical nomenclature is wrong written.

It’s not only to FDG but also to PSMA or [18F]NaF – please provide appropriate changes. We have carefully read the article and have tried hard to convert all uses of radioactive material to the format suggested in the consensus nomenclature rules. However, after first defining the correct radiochemical name, we have referred to these forms with abbreviations such as FDG.

4)    SUVmax is usually written to one decimal place. – this was not changed.

The only place where it was displayed to two decimal places was when we were citing an SUVmax cut-off that was given in the literature. But we have now rounded this off to one decimal place.
